# Analyzing the Sample Complexity of Self-Supervised Image Reconstruction Methods

**Tobit Klug, Dogukan Atik, Reinhard Heckel**
School of Computation, Information and Technology
Technical University of Munich
`tobit.klug, dogukan.atik, reinhard.heckel {@tum.de}`

## Abstract

Supervised training of deep neural networks on pairs of clean image and noisy measurement achieves state-of-the-art performance for many image reconstruction tasks, but such training pairs are difficult to collect. Self-supervised methods enable training based on noisy measurements only, without clean images. In this work, we investigate the cost of self-supervised training in terms of sample complexity for a class of self-supervised methods that enable the computation of unbiased estimates of gradients of the supervised loss, including noise2noise methods. We analytically show that a model trained with such self-supervised training is as good as the same model trained in a supervised fashion, but self-supervised training requires more examples than supervised training. We then study self-supervised denoising and accelerated MRI empirically and characterize the cost of self-supervised training in terms of the number of additional samples required, and find that the performance gap between self-supervised and supervised training vanishes as a function of the training examples, at a problem-dependent rate, as predicted by our theory.

## 1 Introduction

Deep neural networks trained in a supervised fashion to map a noisy measurement to a clean image achieve state-of-the-art performance for image reconstruction tasks including image denoising [48, 12], image super-resolution [8, 26] and accelerated magnetic resonance imaging (MRI) [13, 11].

However, collecting clean training images is sometimes not possible, and is often expensive and time-consuming. For example, to collect clean target images for denoising is difficult since a sensor in a camera only collects noisy images [33]. To collect target images for MRI is difficult as it requires acquiring fully sampled data with long scan times which, introduces difficulties like increased motion artifacts.

This has motivated research on self-supervised methods that enable the training of neural networks from noisy measurements of images only. Self-supervised approaches are generally based on constructing a self-supervised loss. For example, noise2noise [24] constructs a self-supervised loss based on a noisy image and a second noisy observation of the same image. Noisier2noise [30], noise2self [2], and neighbor2neighbor [18] construct losses based on a single noisy image for denoising. One might expect that a model trained with a self-supervised loss computed from noisy images performs worse than the same model trained in a supervised manner. And indeed, for image reconstruction problems a model trained with many of the self-supervised losses (in particular noisier2noise, noise2self, and neighbor2neighbor) performs worse than a model trained with a supervised loss, even when abundant training data is available.

37th Conference on Neural Information Processing Systems (NeurIPS 2023).

However, with infinite training data, noise2noise [24] self-supervised training promises to achieve the performance of supervised training, since the noise2noise loss enables the computation of unbiased estimates of the gradients of the supervised loss.

In this paper, we study a class of self-supervised methods including noise2noise based on constructing unbiased estimates of the gradients of the supervised loss. We characterize the cost of self-supervised training in terms of the number of training examples required to achieve the same performance as supervised training. Our contributions are:

- **Finite sample theory.** We start by viewing the noise2noise loss as enabling computation of unbiased estimates of the gradients of the supervised loss. Based on this view, we characterize the sample complexity of noise2noise like self-supervised training. We find that the risk of a method trained in a noise2noise-like fashion as a function of training examples approaches the optimal risk at the same rate as supervised training (i.e., at rate $1/N$, where $N$ is the number of training examples). However, to reach the same performance as with supervised training more training examples are required. The amount of extra training examples required is a function of the self-supervised loss and of the reconstruction problem.

- **Empirical sample complexity of self-supervised denoising.** We empirically characterize the performance of noise2noise self-supervised training for Gaussian and camera-noise image denoising as a function of the training examples, and find that, as predicted by theory, once the training set size becomes sufficiently large noise2noise training yields a denoiser essentially on par with supervised training. For Gaussian denoising, we also characterize the performance of noisier2noise and neighbor2neighbor like training as a function of the number of training examples, and find, again as expected, models trained with a noisier2noise and neighbor2neighbor self-supervised loss perform worse than models trained in a supervised manner, even when abundant training data is available.

- **Empirical sample complexity of self-supervised compressive sensing.** Finally, we characterize the performance of noise2noise-like self-supervised training similar to the approaches from Yaman et al. [45] and Millard and Chiew [29] for compressive sensing reconstruction as a function of the training examples. Again the performance gap between models trained with a self-supervised and supervised loss goes to zero as a function of the training examples at a problem dependent rate. Perhaps surprisingly, the performance gap is vanishing already for small training set sizes because the gradients constructed from the self-supervised loss have a similar variance than the gradients constructed from the supervised loss.

Together, our results show that networks trained with a self-supervised loss based on computing unbiased estimates of the gradients of the supervised loss perform as well as model trained in a supervised fashion at the cost of additional training examples required.

**Related work.** We consider self-supervised learning methods since they enable training neural networks for problems where clean examples are not available or are scarce. We discuss the most related works throughout the paper. Another training technique that is useful when clean data is scarce are data augmentation techniques [7, 10]. Another class of methods that is interesting for the regime, where no clean data is available are zero-shot methods that are not based on any data, such as deep-image-prior based methods [40, 15, 14, 6]. There is also a variety of zero shot methods that rely partly on self-supervised loss functions, such as Yaman et al. [44]'s method for compressive sensing, noise2fast [25], and zero-shot noise2noise [28].

## 2 Background on self-supervised learning based on estimates of gradients of the supervised loss

Let $f_{\boldsymbol{\theta}} \colon \mathbb{R}^m \to \mathbb{R}^n$ be a neural network with parameters $\boldsymbol{\theta}$ for estimating an image $\mathbf{x} \in \mathbb{R}^n$ based on a measurement $\mathbf{y} \in \mathbb{R}^m$. Our goal is to find a neural network $f_{\boldsymbol{\theta}}$ that has a small risk

$$R(\boldsymbol{\theta}) = \mathbb{E}_{(\mathbf{x},\mathbf{y})}\left[\ell(f_{\boldsymbol{\theta}}(\mathbf{y}), \mathbf{x})\right]. \tag{1}$$

Here, expectation is over the signal-measurement pairs $(\mathbf{x}, \mathbf{y})$ (for example a clean image $\mathbf{x}$ and a noisy measurement $\mathbf{y} = \mathbf{x} + \mathbf{z}$), and $\ell$ is a supervised loss, which we take as the mean-squared error.

Since the data distribution is unknown, we can't compute and minimize the risk. In supervised training, we approximate the risk with an empirical risk computed based on $N$ pairs of training examples. This requires pairs of ground-truth signal $\mathbf{x}$ and associated measurements $\mathbf{y}$.

We consider self-supervised training with a self-supervised loss that, unlike the supervised loss, does not require ground-truth images. Instead, it depends on another, randomized measurement of the ground-truth image, denoted by $\mathbf{y}'$. We are given $N$ pairs of a randomized measurement and original measurement $(\mathbf{y}'_1, \mathbf{y}_1), \ldots, (\mathbf{y}'_N, \mathbf{y}_N)$, and train a network to reconstruct a clean image from the original measurement by minimizing

$$\mathcal{L}_{\text{SS}}(\boldsymbol{\theta}) = \frac{1}{N} \sum_{i=1}^{N} \ell_{\text{SS}}(f_{\boldsymbol{\theta}}(\mathbf{y}_i), \mathbf{y}'_i). \tag{2}$$

We consider loss functions $\ell_{\text{SS}}$, like the noise2noise loss discussed below, with the property that in expectation over the training data, a gradient of the self-supervised loss is also a gradient of the risk $R(\boldsymbol{\theta})$, and thus with sufficiently many training examples, we expect a network trained with such a self-supervised loss to achieve the same performance as when trained in a supervised manner.

However, the 'noise' induced by relying on randomized measurements of the original image instead of relying on the original image increases the variance of the gradients computed from the self-supervised loss. Therefore, self-supervised training requires more training examples to yield a network on par with supervised training. Next, we discuss two self-supervised losses, one for denoising and one for compressive sensing in the context of MRI.

## 2.1 Noise2noise loss for denoising

For denoising, our goal is to train a neural network $f_{\boldsymbol{\theta}}$ to estimate an image $\mathbf{x}$ from a noisy observation $\mathbf{y} = \mathbf{x} + \mathbf{z}$, where $\mathbf{z}$ is additive Gaussian, non-Gaussian, or even structured noise. Noise2noise assumes access to pairs of noisy observations $(\mathbf{y}_1, \mathbf{y}'_1), \ldots, (\mathbf{y}_N, \mathbf{y}'_N)$, where $\mathbf{y}_i = \mathbf{x}_i + \mathbf{z}_i$ and $\mathbf{y}'_i = \mathbf{x}_i + \mathbf{e}_i$. Here, $\mathbf{e}_i$ is zero-mean noise, independent of but not necessarily of the same distribution as the noise $\mathbf{z}_i$. Lehtinen et al. [24] introduced the (noise2noise) self-supervised loss

$$\ell_{\text{SS}}(f_{\boldsymbol{\theta}}(\mathbf{y}), \mathbf{y}') = \|f_{\boldsymbol{\theta}}(\mathbf{y}) - \mathbf{y}'\|_2^2, \tag{3}$$

which aims at finding a network that predicts one noisy observation of an image based on another one, therefore the name noise2noise.

In expectation, a minimizer of the self-supervised loss is also a minimizer of the associated risk (1), as formalized by the proposition below. Thus, training with the self-supervised loss (3) with infinitely many training examples is as good as supervised training with infinitely many training examples. The proof is in Appendix A.

**Proposition 1.** *Suppose that a signal $\mathbf{x}$ and a corresponding measurement $\mathbf{y}$ are drawn from a joint distribution, and let $\mathbf{y}' = \mathbf{x} + \mathbf{e}$ be another randomized measurement of the signal. Assume that the noise $\mathbf{e}$ is uncorrelated with the residual, i.e., $\mathbb{E}_{(\mathbf{x},\mathbf{y},\mathbf{e})} \left[ (f_{\boldsymbol{\theta}}(\mathbf{y}) - \mathbf{x})^T \mathbf{e} \right] = 0$, for all $\boldsymbol{\theta}$. Then, the minimizer $\boldsymbol{\theta}$ of the self-supervised risk $\mathbb{E}_{(\mathbf{y},\mathbf{y}')} \left[ \|f_{\boldsymbol{\theta}}(\mathbf{y}) - \mathbf{y}'\|_2^2 \right]$ is also a minimizer of the supervised risk $\mathbb{E}_{(\mathbf{x},\mathbf{y})} \left[ \|f_{\boldsymbol{\theta}}(\mathbf{y}) - \mathbf{x}\|_2^2 \right]$.*

Noise2noise self-supervised training has been applied to many domains including biomedical imaging [5, 3, 19], channel estimation [51] or acoustic sensing [23].

## 2.2 Compressive sensing

For accelerated MRI, our goal is to train a network $f_{\boldsymbol{\theta}}$ to reconstruct an image $\mathbf{x}$ from an undersampled measurement $\mathbf{y} = \mathbf{MFx}$ in the frequency domain. Here, $\mathbf{M} \in \mathbb{R}^{n \times n}$ is an undersampling mask (i.e., a diagonal matrix with zeros and ones on its diagonal), and $\mathbf{F} \in \mathbb{C}^{n \times n}$ is the Fourier transform.

We do not have access to a ground-truth image $\mathbf{x}$. Instead, we are given an undersampled measurement $\mathbf{y} = \mathbf{MFx}$ as well as a second randomized measurement of the same image, obtained as $\mathbf{y}' = \mathbf{M}'\mathbf{Fx}$, where $\mathbf{M}' \in \mathbb{R}^{n \times n}$ is a randomized mask which is zero or one on its diagonal, and is one with

non-zero probability, so that $\mathbf{W} = \mathbb{E}\left[\mathbf{M}'\right]^{-1/2}$ exists. From this measurement, we can compute a self-supervised loss defined as

$$\ell_{\mathrm{SS}}(f_{\boldsymbol{\theta}}(\mathbf{y}), \mathbf{y}') = \|\mathbf{W}(\mathbf{M}'\mathbf{F}f_{\boldsymbol{\theta}}(\mathbf{y}) - \mathbf{y}')\|_2^2, \tag{4}$$

where $\mathbf{W} = \mathbb{E}\left[\mathbf{M}'\right]^{-1/2}$ is a diagonal weighting mask. The loss is evaluated only on the frequencies that are given through the randomized mask $\mathbf{M}'$ and are weighted by how often a frequency occurs in the randomized mask $\mathbf{M}'$ (thus, the multiplication with the weighting mask $\mathbf{W}$).

Analogous as for the noise2noise loss for denoising, in expectation, a minimizer of the self-supervised loss is also a minimizer of the associated risk, as formalized by the following proposition. The proof is in Appendix A.

**Proposition 2.** *Suppose that a signal* $\mathbf{x}$ *and a corresponding measurement* $\mathbf{y}$ *are drawn from some joint distribution (e.g., as* $\mathbf{y} = \mathbf{MFx}$*, where* $\mathbf{M}$ *is a mask), and let* $\mathbf{y}' = \mathbf{M}'\mathbf{Fx}$ *be an independent measurement taken with a randomized mask* $\mathbf{M}'$ *with 0's or 1's on it's diagonal and non-zero probability of a one on the diagonal. Then a minimizer* $\boldsymbol{\theta}$ *of the self-supervised risk* $\mathbb{E}_{(\mathbf{y},\mathbf{y}')}\left[\|\mathbf{W}(\mathbf{M}'\mathbf{F}f_{\boldsymbol{\theta}}(\mathbf{y}) - \mathbf{y}')\|_2^2\right]$ *with* $\mathbf{W} = \mathbb{E}\left[\mathbf{M}'\right]^{-1/2}$ *is also a minimizer of the supervised risk* $\mathbb{E}_{(\mathbf{x},\mathbf{y})}\left[\|f_{\boldsymbol{\theta}}(\mathbf{y}) - \mathbf{x}\|_2^2\right]$.

A variety of works consider training a neural network with a self-supervised loss similar to (4) [24, 45, 42, 17, 16, 52, 29]. However, none of those works studied the sample complexity. Further, Lehtinen et al. [24] train with a noise2noise loss, but sample new masks $\mathbf{M}$ and $\mathbf{M}'$ in every training epoch, which requires access to fully sampled measurements. Similar to our setup, Yaman et al. [45] and Millard and Chiew [29] fix the set of undersampled measurements per image available for creating the masks $\mathbf{M}$ and $\mathbf{M}'$. However, Yaman et al. [45] and Millard and Chiew [29] construct a noiser2noise-like self-supervised loss, i.e., the input mask $\mathbf{M}$ implements a higher undersampling factor on the training inputs than the undersampling factor used at inference. We assume the mask $\mathbf{M}$ to have the same distribution at training and inference, which is important for Proposition 2 to hold.

## 3    Finite sample theory for self-supervised denoising

We start by studying the performance of a denoiser trained with a self-supervised noise2noise loss analytically in the finite sample regime. We measure performance in terms of the risk $R(\boldsymbol{\theta}) = \mathbb{E}\left[\|f_{\boldsymbol{\theta}}(\mathbf{y}) - \mathbf{x}\|_2^2\right]$ and learn an estimator $f_{\boldsymbol{\theta}}$ by minimizing the self-supervised objective function (2), where we take the noise2noise loss (3) as the self-supervised loss.

**Finite-sample risk bound for linear denoising.**    We consider a simple linear denoising problem, where the joint distribution of the image and corresponding measurement is as follows. The signal $\mathbf{x}$ is drawn from a $d$-dimensional linear subspace according to $\mathbf{x} = \mathbf{Uc}$, where $\mathbf{U} \in \mathbb{R}^{n \times d}$ is an orthonormal basis for the subspace, and where $\mathbf{c} \sim \mathcal{N}(0, 1/d\mathbf{I})$. Thus, for large $d$, the vector is drawn approximately uniformly from the intersection of the subspace with the unit sphere. We draw a measurement as $\mathbf{y} = \mathbf{x} + \mathbf{z}$, where $\mathbf{z} \sim \mathcal{N}(0, \sigma_z^2/n\mathbf{I})$ is Gaussian noise, and then draw a second measurement as $\mathbf{y}' = \mathbf{x} + \mathbf{e}$, where $\mathbf{e} \sim \mathcal{N}(0, \sigma_e^2/n\mathbf{I})$ is the target noise. With this scaling, the expected SNR of the denoising problem is $1/\sigma_z^2$.

We consider a linear estimator of the form $f_{\mathbf{W}}(\mathbf{y}) = \mathbf{Wy}$. Note that the optimal linear estimator (i.e., the estimator that minimizes the risk) is $\mathbf{W}^* = \frac{1}{1+\sigma_z^2\frac{d}{n}}\mathbf{UU}^T$. The optimal estimator projects the measurement onto the subspace and shrinks the projected measurement by a noise-dependent factor.

We provide a bound on the expected risk of the estimator that is learned by minimizing the self-supervised loss function (2) by running the stochastic gradient method for one epoch. Given a dataset $\mathcal{D} = \{(\mathbf{y}_1, \mathbf{y}_1'), \ldots, (\mathbf{y}_N, \mathbf{y}_N')\}$ consisting of pairs of two noisy measurements $\mathbf{y}_i = \mathbf{x}_i + \mathbf{z}_i, \mathbf{y}_i' = \mathbf{x}_i + \mathbf{e}_i$ drawn i.i.d. from the distribution specified above, we start the stochastic gradient method at $\mathbf{W}_0 = 0$ and update

$$\mathbf{W}_{k+1} = \mathbf{W}_k - \alpha_k \nabla_{\mathbf{W}}\|\mathbf{W}\mathbf{y}_k - \mathbf{y}_k'\|_2^2,$$

for $k = 1, \ldots, N$. Here, $\alpha_k$ is the stepsize.

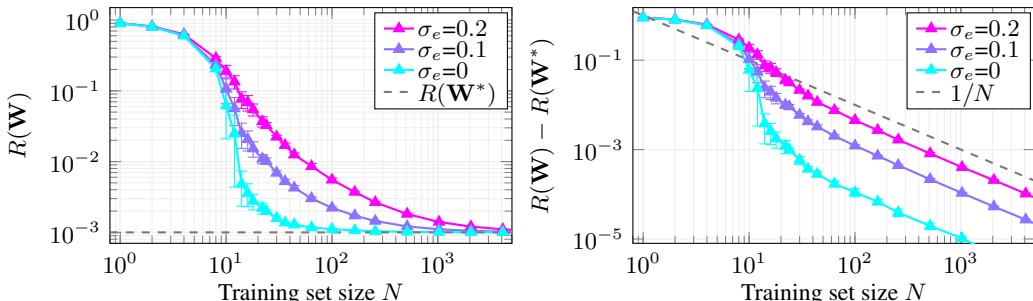

Figure 1: **Subspace denoising.** Simulated risk of a linear estimator learned with early stopped gradient descent for different levels of target noise $\sigma_e$, signal dimension $d = 10$, ambient dimension $n = 100$, and input noise level $\sigma_z = 0.1$. **Left:** The larger the target noise level, the more training examples are required for self-supervised training to approach the optimal risk $R(\mathbf{W}^*)$. **Right:** Plotting the risk minus the optimal risk reveals the convergence rate to be $1/N$. Error bars show the standard deviation over 5 independent runs.

**Theorem 1.** *Consider the estimate $\mathbf{W}_N$ obtained by running the SGM for $N$ iterations on the training set $\mathcal{D}$ with a decaying stepsize $\alpha_k = \frac{1}{c+k}$, where $c$ is a constant. Then, the expected generalization error, where expectation is over the random training set $\mathcal{D}$, obeys*

$$\mathbb{E}\left[R(\mathbf{W}_N))\right] \leq R(\mathbf{W}^*) + \frac{1/d + \sigma_z^2/n}{(\sigma_z^2/n)^2} \frac{1}{N-2}\left(2 + \left(12\sigma_z^2\frac{d}{n} + \sigma_e^2(1+\sigma_z^2)\right)^2\right). \quad (5)$$

The proof (detailed in Appendix B), is based on an standard convergence analysis of the stochastic gradient method [32, 43, 35]. In a nutshell, from the noise2noise self-supervised loss we compute stochastic gradients of the risk as $\nabla \ell_{\mathrm{SS}}(f_\theta(\mathbf{y}_i), \mathbf{y}_i')$, where $(\mathbf{y}_i, \mathbf{y}_i')$ is a training example. This stochastic gradient is an unbiased estimate of the gradient of the risk, $\nabla R(\theta)$. The variance of the stochastic gradient determines the rate on the RHS of equation (5).

Theorem 1 establishes that the risk is upper bounded by a problem-dependent noise floor, which is the risk of the optimal estimator, $R(\mathbf{W}^*)$, plus a term associated with minimizing a self-supervised loss constructed from finitely many training examples. This term decays as $c/N$, which is exactly the rate we see in the simulations below for this setup. Moreover, the term associated with minimizing the self-supervised loss becomes larger in the noise variance $\sigma_e^2$, and thus reflects that the self-supervised loss is a worse approximation of the supervised loss, as $\sigma_e^2$ increases.

**Numerical results for linear denoising.** Figure 1 shows the risk of the linear denoiser $\mathbf{W}$ trained on $N$ examples from the linear subspace model. The linear estimator is learned by applying gradient descent to the self-supervised loss function (2) regularized with early stopping. As predicted by Theorem 1, the performance of the linear estimator $\mathbf{W}$ converges to the performance of the optimal estimator $\mathbf{W}^*$ at the rate $1/N$ (right plot in Figure 1). Also as predicted by the Theorem, larger levels of target noise $\sigma_e$ require more training examples, reflected by the term that multiplies with $1/N$ increasing in $\sigma_e^2$.

## 4 Empirical results for self-supervised denoising

In this section, we study the performance of a network trained for denoising with the self-supervised objective function (2) and noise2noise loss (3) as a function of the number of training examples, and compare to the performance of the network trained in a supervised fashion. We consider Gaussian denoising and denosing real-world-camera noise.

Throughout this section, we focus on training a U-net [36] with 7.4M network parameters. A U-net is a good choice, since it is widely used and gives near SOTA models for image denoising [4, 12, 50, 49]. However, our qualitative results are independent of the network architecture. In the appendix, we present results for SwinIR [26], an attention-based SOTA architecture, confirming this.

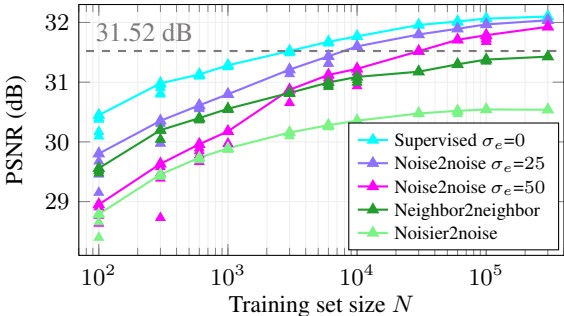
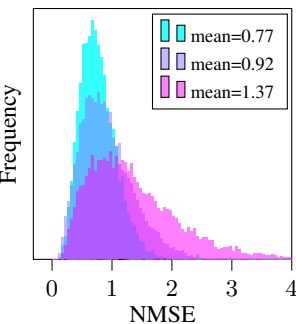

Figure 2: **Gaussian image denoising. Left:** Noise2noise training approaches the performance of supervised training as the number of training images increases at a rate dependent on the target noise level $\sigma_e$. Networks trained with noisier2noise and neighbor2neighbor also improve as a function of the training examples, but are far from approaching the performance of networks trained in a supervised fashion. **Right:** Histogram of the variance of the stochastic gradients. The noisier the gradients, the slower the convergence as a function of the training examples, $N$.

## 4.1 Gaussian denoising

**Setup.** We consider denoising images $\mathbf{x}$ from noisy measurements $\mathbf{y} = \mathbf{x} + \mathbf{z}$ with noise level $\sigma_z = 25$ (for pixel values in $[0, 255]$). For the noisy targets $\mathbf{y}' = \mathbf{x} + \mathbf{e}$, we study noise levels $\sigma_e \in \{0, 25, 50\}$, where $\sigma_e = 0$ corresponds to supervised training. The target noise level determines how fast as a function of the training set size self-supervised approaches the performance of supervised training. In all our experiments, the measurement noise $\mathbf{z}_i$ and target noise $\mathbf{e}_i$ are only sampled once per clean image $\mathbf{x}_i$ during training, which corresponds to the practical setup in which we are given two noisy measurements per image. We use cropped patches of size $128 \times 128$ from ImageNet [37] to create training sets with 100 to 300k images. We pick the best run out of different runs with independently sampled network initialization. See Appendix C.1.2 for training details.

**Results and discussion.** Figure 2 shows the performance in PSNR of U-nets trained in a supervised and self-supervised manner for different training set sizes $N$. We find, as expected from the theory in Section 3, that the gap between noise2noise like self-supervised training and supervised training vanishes as a function of the training examples, at a rate that increases as $\sigma_e$ decreases.

At 100 training images the performance gap between supervised and self-supervised training with $\sigma_e = 25$ ($\sigma_e = 50$) is 0.645dB (1.497dB), and at 100k training images the gap is reduced to 0.097dB (0.277 dB). For this specific setup, we can read of the number of additional images required to meet a certain denoising performance relative to what can be achieved with supervised training. For example, to achieve the performance of supervised training on 3k images, self-supervised training with noise on the training targets $\sigma_e = 25$ ($\sigma_e = 50$) requires 10k (30k) images (gray dashed line in Figure 2).

The theory in Section 3 is based on a convergence analysis of the stochastic gradient method, and the rate at which performance improves as a function of the number of training examples, $N$, is determined by how well the stochastic gradients of the risk, $\nabla \ell_{\mathrm{SS}}(f_{\boldsymbol{\theta}}(\mathbf{y}_i), \mathbf{y}'_i)$ (where $(\mathbf{y}_i, \mathbf{y}'_i)$ is a training example), approximate the gradient of the risk, i.e., $\nabla R(\boldsymbol{\theta})$. In Figure 2 (right panel) we show the histograms of the normalized variance of stochastic gradients, i.e., normalized estimates of the MSE $\|\nabla \ell_{\mathrm{SS}}(f_{\boldsymbol{\theta}}(\mathbf{y}_i), \mathbf{y}'_i) - \nabla R(\boldsymbol{\theta})\|_2^2$ after one epoch of training; see Appendix E for details. It can be seen that the larger the variance of the stochastic gradients, the slower the improvement as a function of the training examples, $N$.

Self-supervised training with noisier2noise and neighbor2neighbor in general does not yield networks as good as trained in a supervised fashion, even if abundant training data is available. Noisier2noise [30] and neighbor2neighbor [18] only require a single noisy measurement per clean image for training. However, noisier2noise adds additional noise to the training inputs, which introduces a mismatch between the problem solved during training and inference. Neighbor2neighbor relies on assumptions on the similarity between neighboring pixels in the clean image that only hold approximately in practice. In Figure 2 we see that noisier2noise and neighbor2neighbor training,

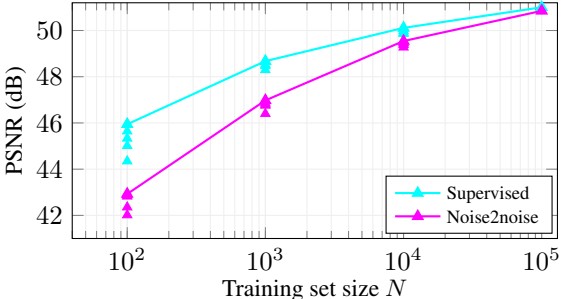 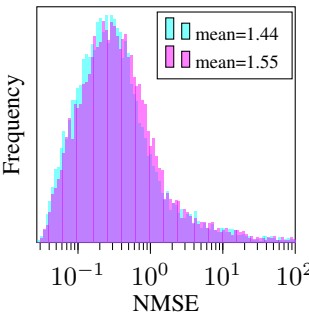

Figure 3: **Real-world camera noise denoising on SIDD**. **Left:** Similar to our results for Gaussian denoising a model trained with noisy targets in a noise2noise manner approaches the performance of a model trained with ground truth targets in a supervised manner as the number of training patches gets large. **Right:** Histogram of the variance of the stochastic gradients. The stochastic gradients of the self-supervised loss are slightly noisier than the ones of the supervised loss.

unlike noise2noise, yield methods that perform worse than methods trained in a supervised fashion even if abundant training data is available. For training details see Appendices C.1.3 and C.1.4.

For the results in Figure 2 we consider a U-net of fixed size. The performance of a network trained in a supervised and self-supervised manner depends on the network architecture and size. However, since we investigate training schemes, we expect our qualitative findings for relative performance differences to continue to hold for other architectures and network sizes. In Appendix C.1.1 we provide results for U-nets of varying sizes and in Appendix C.1.5 we provide results for SwinIR [26], a recent state-of-the-art network for image denoising. The results are as expected analogous to the ones for the U-net presented here.

### 4.2 Real-world camera noise

We now study self-supervised denoising for real-world camera noise from the Smartphone Image Denoising Dataset (SIDD) [1]. The noise in the SID dataset is structured and not Gaussian.

**Setup.** We train U-nets on increasing amounts of patches of size $128 \times 128$ from the training set, validate on the first 10 scenes, and report the test performance on the remaining 30 scenes from the validation set. The dataset consists of 150 noisy images per scene of which 2 are used as input and target for self-supervised training. Ground truth images for supervised training are estimated from all 150 noisy images. Training and testing is done in the raw-RGB space; see Appendix C.2 for details.

**Results and discussion.** In Figure 3 we see that performance improves steeper as a function of the training set size compared to Gaussian denoising in Figure 2, which we attribute to the high variety of noise levels and lighting conditions present in the SID dataset. However, analogous to the Gaussian case the model trained in a self-supervised manner closely approaches the performance of the model trained in a supervised manner as the number of training examples grows large.

## 5 Empirical results for self-supervised compressive sensing

We now study the performance as a function of the training set size of a network trained for compressive sensing reconstruction by minimizing the self-supervised objective (2) with the noise2noise-like loss (4), relative to the performance achieved by the same network with supervised training.

We perform two sets of experiments: First we experiment on natural images obtained from ImageNet, since this allows us to explore the compressive sensing problem over a wider range of training set sizes than existing medical image datasets allow. Then, we investigate accelerated MRI reconstruction on real world multi-coil measurements from the fastMRI dataset [46]. We consider a U-net in the main body, but again our qualitative results are independent of the network, and we demonstrate this with experiments on the VarNet, a state-of-the-art architecture, in the appendix.

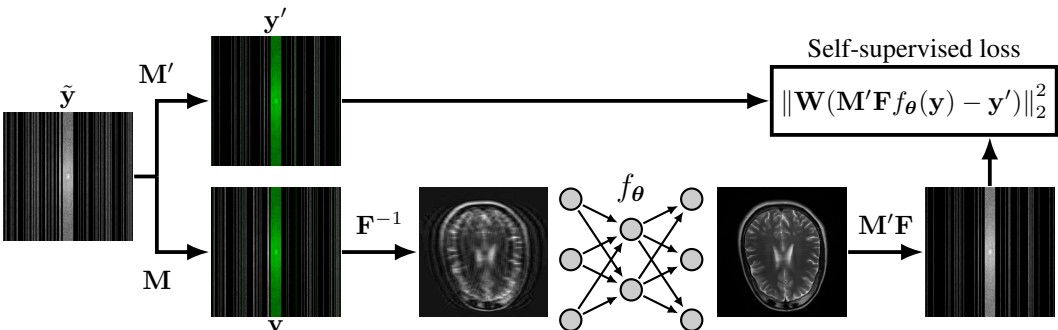

Figure 4: **Self-supervised training scheme for compressive sensing MRI.** The overlap between constructed measurements $\mathbf{y}, \mathbf{y}'$ is indicated in green and stems from both masks sampling center frequencies with probability 1 and an expected overlap of non-center frequencies of size $p'q$. The scheme depicts single-coil measurements for conciseness.

**Sampling scheme.** The setup from Section 2.2 requires a pair of source measurement $\mathbf{y}$ (that can be arbitrarily obtained) and a target measurement that is sampled independently of the source measurement as $\mathbf{y}' = \mathbf{M}'\mathbf{F}\mathbf{x}$, where $\mathbf{M}'$ is a random mask with a non-zero probability of selecting each frequency. While in clinical MRI protocols, we can acquire two independently undersampled measurements per image, we typically acquire only one. We therefore slightly adapt our sampling scheme as follows to generate two measurements $\mathbf{y}$ and $\mathbf{y}'$ from one undersampled measurement $\tilde{\mathbf{y}}$ that is undersampled with factor $\mu$. See Figure 4 for an illustration of the scheme.

Our goal is to train a network to perform reconstruction based on a measurement $\mathbf{y}$ undersampled with factor $p < \mu$. We consider Cartesian undersampling, where the frequency domain (also called k-space) is sampled column-wise. Since the low-frequency components of an image contain the largest portion of the signal energy, we follow the common practice to sample a fraction $\nu = 0.08$ of the center columns in the k-space with probability 1, and assign the center frequencies to both measurements $\mathbf{y}$ and $\mathbf{y}'$. We then add a random fraction $\frac{p-\nu}{\mu-\nu}$ of the non-center columns in $\tilde{\mathbf{y}}$ to $\mathbf{y}$, which is now a measurement undersampled with factor $p$ by design. The target $\mathbf{y}'$ is constructed by adding the remaining frequencies that are in $\tilde{\mathbf{y}}$ but not in $\mathbf{y}$, and an additional fraction $p'q$ of the non-center columns in $\mathbf{y}$, where $p' = \frac{p-\nu}{1-\nu}$. The rationale behind this approach is that if the masks $\mathbf{M}$ and $\mathbf{M}'$ sample fractions $p'$ and $q$ independently from all non-center frequencies, then the expected overlap of frequencies selected by both masks is $p'q$ with $q = \frac{\mu-p}{1-p}$.

We compute the diagonal weight mask from equation (4) as $\mathbf{W} = \mathbb{E}\left[\mathbf{M}'\right]^{-1/2}$, which yields 1's at entries corresponding to center frequencies and $1/\sqrt{q}$ otherwise. In our experiments we set $\nu = 0.08$, $p = 0.25$ and $\mu \in \{0.28, 0.33\}$ resulting in $q \in \{0.05, 0.11\}$.

## 5.1 Compressive sensing for natural images

We start with compressive sensing experiments on natural images, because the largest existing MRI dataset for image reconstruction research (fastMRI) only has 55k images, and we want to vary the training set size far beyond that. In the next section we conduct experiments on fastMRI.

**Setup.** We use cropped images of size $100 \times 100$ from ImageNet [37] as ground truth images $\mathbf{x}$ and obtain undersampled complex-valued measurements in the frequency domain as $\mathbf{y} = \mathbf{M}\mathbf{F}\mathbf{x}$ and $\mathbf{y}' = \mathbf{M}'\mathbf{F}\mathbf{x}$. We draw random training sets with 50 to 1M patches from a fixed pool of 1M images.

For all training set sizes, we train a U-net with about 1M network parameters and pick the best run out of different runs with independently sampled training set and network initialization. The network takes as input a coarse reconstruction obtained as $\mathbf{F}^{-1}\mathbf{y}$, where $\mathbf{y}$ is the zero-filled measurement (i.e., $f_{\boldsymbol{\theta}}(\mathbf{y}) = \text{U-net}_{\boldsymbol{\theta}}(\mathbf{F}^{-1}\mathbf{y})$. The measurements are complex-valued, and we take one channel of the U-net as the imaginary part and one as the real part. The reconstructed image is obtained from the complex-valued network output by taking the absolute value. See Appendix D.1 for training details.

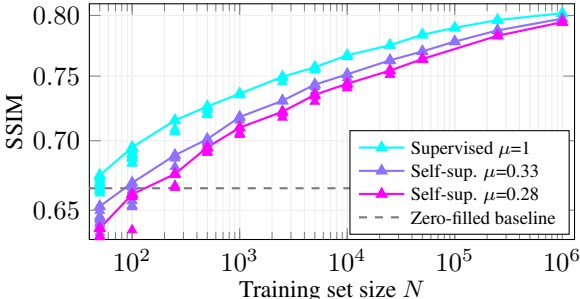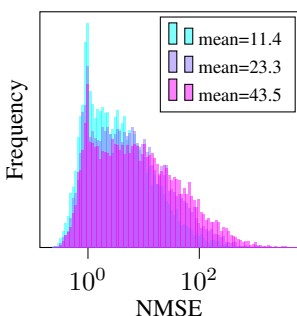

Figure 5: **Compressive sensing for natural images. Left:** The larger the fraction of frequencies in the target, $\mu$, the faster the network's performance improves as a function of the number of training examples, $N$. As the training set size becomes large, self-supervised performance approaches supervised performance. **Right:** Histogram of the variance of the stochastic gradients. The noisier the gradients (i.e., the smaller $\mu$), the slower the convergence as a function of the training examples.

**Results and discussion.** Figure 5 shows the performance in SSIM as a function of the training set size for supervised and self-supervised training with fractions $\mu = 0.33$ and $\mu = 0.28$ of all frequencies available for self-supervised training. Note that the fraction of frequencies in the target plays the same role as the noise variance for noise2noise denoising: The larger the fraction, the better in general. Since we keep the undersampling factor $p = 0.25$ of the training inputs $\mathbf{y}$ fixed, a smaller $\mu$ implies a smaller undersampling factor $q$ (less frequencies) of the training targets $\mathbf{y}'$.

Networks trained in a self-supervised manner with sufficiently many training examples perform as well as the same network trained in a supervised manner (but with fewer examples). The fewer measurements are available, i.e., the smaller $\mu$, the more training examples are required to achieve the same performance as the same network trained in a supervised manner. This parallels our theoretical and empirical findings for denoising where noise2noise training approaches the performance of supervised training as a function of the training set size and at a rate dependent of the training targets.

Figure 5, right panel, depicts histograms of the variance of the stochastic gradients. The noisier the gradients, the slower the performance increase as a function of the training examples.

The performance of the zero-filled baseline in Figure 5 is computed between the network inputs $\mathbf{F}^{-1}\mathbf{y}$ and ground truth images $\mathbf{x}$ in the test set. As we can see self-supervised training requires at least 250 training images to outperform this trivial baseline.

### 5.2 Compressive sensing MRI

Next, we discuss the performance of noise2noise self-supervised and supervised training for multi-coil MRI reconstruction. MRI scanners use parallel imaging, where a single scan simultaneously collects measurements $\mathbf{y}_1, \ldots, \mathbf{y}_C$ of the same object with $C$ coils, where $\mathbf{y}_j = \mathbf{MFS}_j\mathbf{x}$ with complex valued images $\mathbf{x} \in \mathbb{C}^n$ and diagonal coil specific sensitivity maps $\mathbf{S}_j \in \mathbb{C}^{n \times n}$, which we estimate from the center of the k-space using ESPIRiT [39].

**Setup.** We conduct our experiments on a subset of the fastMRI brain dataset [46]. We design fixed training sets with 50 to 50k images. We train U-nets with 31M network parameters. To demonstrate that our results for comparing supervised and self-supervised training schemes translate qualitatively to other network architectures, we further train end-to-end variational networks (E2E-VarNet) [38], a state-of-the-art architecture, of size 9M parameters. We pick the best run out of different runs with independently sampled network initialization. See Appendix D.2 for training details.

The network inputs are computed through SENSE-1 coil combination [34] of the coil measurements as $\sum_{j=1}^{C} \mathbf{S}_j^* \mathbf{F}^{-1} \mathbf{y}_j$, where $\mathbf{S}_j^*$ denotes the complex conjugate of the sensitivity map. Ground truth images for evaluation are computed analogously as $|\sum_{j=1}^{C} \mathbf{S}_j^* \mathbf{F}^{-1} \mathbf{y}_{j,\text{full}}|$, where $\mathbf{y}_{j,\text{full}}$ is the fully sampled coil k-space provided by the fastMRI dataset. To compute the training loss in the k-space between given and predicted coil measurements, the set of predicted coil measurements is obtained from the network output as $\mathbf{FS}_1 f_{\boldsymbol{\theta}}(\mathbf{y}), \ldots, \mathbf{FS}_C f_{\boldsymbol{\theta}}(\mathbf{y})$.

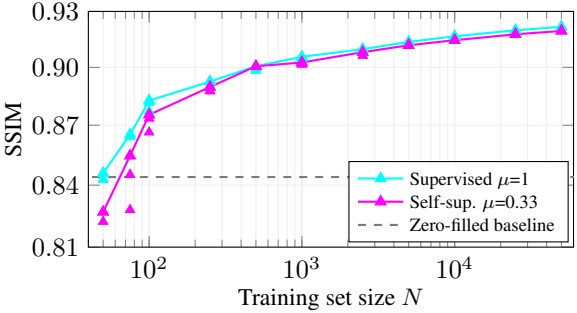
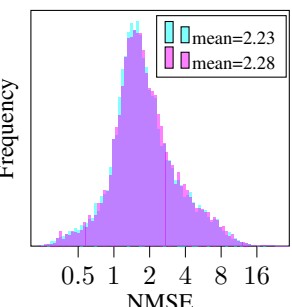

Figure 6: **Compressive sensing MRI. Left:** Already at small training set sizes the performance gap between supervised and self-supervised training vanishes. **Right:** The small performance gap on the left is explained by a small difference in the variances of the supervised and self-supervised gradients.

We follow Yaman et al. [45] and make best use of the given data by re-sampling the input mask $\mathbf{M}$ in every epoch for supervised training. For self-supervised training the given frequencies $\tilde{\mathbf{y}}$ are fixed, but we can re-sample the split into measurements $\mathbf{y}, \mathbf{y}'$. As the network sees more combinations of input/target pairs, the performance of supervised and self-supervised training improves.

**Results and discussion.** Figure 6 shows the performance of U-nets trained in a supervised and self-supervised way for multi-coil MRI as a function of the training set size. We report the performance in SSIM, since a high score in SSIM was shown to correlate well with a high rating by radiologists in the fastMRI challenge [31]. We find that already for small training set sizes the performance gap between supervised and self-supervised training vanishes to approximately 0.002 in SSIM.

Figure 6, right panel, depicts the histogram of the variance of the stochastic gradients. The difference in variance of the supervised and self-supervised gradients is very small, which explains the small performance gap between supervised and self-supervised training for this setup. The performance of the zero-filled baseline in Figure 6 is computed between the magnitude of the network inputs $|\sum_{j=1}^{C} \mathbf{S}_j^* \mathbf{F}^{-1} \mathbf{y}_j|$ and ground truth images in the test set.

In Appendix D.2.2 we provide additional results for the E2E-VarNet. As expected, the VarNet outperforms the U-net by a significant margin for every training set size, especially for small training set sizes. However, the relative performance gap between VarNets trained in a supervised and self-supervised way is very small, analogous as for the U-net.

# 6 Conclusion

We characterized the sample complexity of deep learning based image reconstruction models trained with a class of self-supervised losses including noise2noise based on computing unbiased estimates of the gradients of the supervised loss. Our work shows that if sufficiently many training examples are available, it is as good to work with pairs of independent measurements as with training data with ground-truth images. In particular for accelerated MRI, training networks only based on undersampled data is sufficient for peak performance.

One advantage of supervised training over self-supervised training that is not reflected in our results is that supervised training has more flexibility in choosing the loss function, and choosing a loss function such as SSIM might give better visual image quality than training with the MSE. Our results of self-supervised training approaching supervised training pertain to training with the MSE.

We hasten to note that the class of methods that enable the computation of unbiased estimates of gradients of the supervised loss studied in this work, are in practice limited to problem setups in which obtaining two independent measurements of the same underlying signal is feasible.

**Reproducibility** The repository at `https://github.com/MLI-lab/sample_complexity_ss_recon` contains the code to reproduce all results in the main body of this paper.

## Acknowledgments

The authors are supported by the Institute of Advanced Studies at the Technical University of Munich, the Deutsche Forschungsgemeinschaft (DFG, German Research Foundation) - 456465471, 464123524, the DAAD, the German Federal Ministry of Education and Research, and the Bavarian State Ministry for Science and the Arts. The authors also acknowledge the financial support by the Federal Ministry of Education and Research of Germany in the programme of "Souveraen. Digital. Vernetzt.". Joint project 6G-life, project identification number: 16KISK002.

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

# A    Proofs for propositions in Section 2

*Proof of Proposition 1.* We have that

$$\mathbb{E}_{\mathbf{x},\mathbf{y},\mathbf{y}'}\left[\|f_{\boldsymbol{\theta}}(\mathbf{y}) - \mathbf{y}'\|_2^2\right] = \mathbb{E}_{\mathbf{x},\mathbf{y},\mathbf{e}}\left[\|f_{\boldsymbol{\theta}}(\mathbf{y}) - \mathbf{x} - \mathbf{e}\|_2^2\right]$$

$$= \mathbb{E}_{\mathbf{x},\mathbf{y}}\left[\|f_{\boldsymbol{\theta}}(\mathbf{y}) - \mathbf{x}\|_2^2\right] - \mathbb{E}_{\mathbf{x},\mathbf{y},\mathbf{e}}\left[(f_{\boldsymbol{\theta}}(\mathbf{y}) - \mathbf{x})^T\mathbf{e}\right] + \mathbb{E}\left[\|\mathbf{e}\|_2^2\right]$$

$$\overset{\text{(i)}}{=} \mathbb{E}_{(\mathbf{x},\mathbf{y})}\left[\|f_{\boldsymbol{\theta}}(\mathbf{y}) - \mathbf{x}\|_2^2\right] + \mathbb{E}\left[\|\mathbf{e}\|_2^2\right],$$

where equation (i) follows from condition in Proposition 1. Noting that the expectation over the norm of the random variable $\mathbf{e}$ above is constant as a function of $\boldsymbol{\theta}$ proves the claim.  $\square$

*Proof of proposition 2.* We show that

$$\mathbb{E}_{\mathbf{M}'}\left[\|\mathbf{W}(\mathbf{M}'\mathbf{F}f_{\boldsymbol{\theta}}(\mathbf{y}) - \mathbf{y}')\|_2^2\right] = \|f_{\boldsymbol{\theta}}(\mathbf{y}) - \mathbf{x}\|_2^2.$$

To see this, note that

$$\mathbb{E}_{\mathbf{M}'}\left[\|\mathbf{W}(\mathbf{M}'\mathbf{F}f_{\boldsymbol{\theta}}(\mathbf{y}) - \mathbf{y}')\|_2^2\right] = \mathbb{E}_{\mathbf{M}'}\left[\|\mathbf{W}(\mathbf{M}'\mathbf{F}f_{\boldsymbol{\theta}}(\mathbf{y}) - \mathbf{M}'\mathbf{F}\mathbf{x})\|_2^2\right]$$

$$= \mathbb{E}_{\mathbf{M}'}\left[\|\mathbf{W}\mathbf{M}'\mathbf{F}(f_{\boldsymbol{\theta}}(\mathbf{y}) - \mathbf{x})\|_2^2\right]$$

$$\overset{\text{(i)}}{=} \mathbb{E}_{\mathbf{M}'}\left[\sum_j w_j^2 m_j^2 [\mathbf{F}(f_{\boldsymbol{\theta}}(\mathbf{y}) - \mathbf{x})]_j^2\right]$$

$$\overset{\text{(ii)}}{=} \sum_j [\mathbf{F}(f_{\boldsymbol{\theta}}(\mathbf{y}) - \mathbf{x})]_j^2$$

$$= \|f_{\boldsymbol{\theta}}(\mathbf{y}) - \mathbf{x}\|_2^2,$$

where equation (i) follows from $\mathbf{M}'$ and $\mathbf{W}$ being diagonal matrices and equation (ii) from $\mathbf{W} = \mathbb{E}\left[\mathbf{M}'\right]^{-1/2}$ the condition in Proposition 2.  $\square$

# B    Proof of Theorem 1 in Section 3

Our result is based on a relatively standard analysis of the convergence of the stochastic gradient method. In order to minimize the loss, we perform one pass of the stochastic gradient method, i.e., starting at $\boldsymbol{\theta}_0 = 0$, we compute the iterations

$$\boldsymbol{\theta}_{k+1} = \boldsymbol{\theta}_k - \alpha_k G_k(\boldsymbol{\theta}_k), \quad k = 1, \ldots, N,$$

where, for notational convenience, we defined the stochastic gradient

$$G_k(\boldsymbol{\theta}) = \nabla_{\boldsymbol{\theta}}\|f_{\boldsymbol{\theta}}(\mathbf{y}_k) - \mathbf{y}_k'\|_2^2, \tag{6}$$

where the pair $(\mathbf{y}_k, \mathbf{y}_k')$ is chosen from the data distribution, independently over $k$.

The stochastic gradient is

$$G(\boldsymbol{\theta}) = \nabla_{\boldsymbol{\theta}}\|f_{\boldsymbol{\theta}}(\mathbf{y}) - \mathbf{y}'\|_2^2$$

$$= \nabla_{\boldsymbol{\theta}}\|f_{\boldsymbol{\theta}}(\mathbf{y}) - (\mathbf{x} + \mathbf{e})\|_2^2$$

$$= \nabla_{\boldsymbol{\theta}}\left(\|f_{\boldsymbol{\theta}}(\mathbf{y}) - \mathbf{x}\|_2^2 - 2\langle f_{\boldsymbol{\theta}}(\mathbf{y}) - \mathbf{x}, \mathbf{e}\rangle + \|\mathbf{e}\|_2^2\right)$$

$$= \nabla_{\boldsymbol{\theta}}\frac{1}{2}\|f_{\boldsymbol{\theta}}(\mathbf{y}) - \mathbf{x}\|_2^2 - \mathbf{J}_{\boldsymbol{\theta},\mathbf{y}}^T\mathbf{e},$$

where $\mathbf{J}_{\boldsymbol{\theta},\mathbf{y}} \in \mathbb{R}^{n \times p}$ is the Jacobian of the function $f_{\boldsymbol{\theta}}$ with respect to the parameters $\boldsymbol{\theta}$. Note that the function $f_{\boldsymbol{\theta}}$ has parameters $\boldsymbol{\theta} \in \mathbb{R}^p$ and maps a vector in $\mathbb{R}^n$ to another vector in $\mathbb{R}^n$; we consider the

Jacobian with respect to the parameters, not with respect to the input vector. The stochastic gradient is an unbiased estimate of the gradient of the risk, i.e.,

$$\mathbb{E}\left[G(\mathbf{W})\right] = \nabla_{\boldsymbol{\theta}}\mathbb{E}\left[\|f_{\boldsymbol{\theta}}(\mathbf{y}) - \mathbf{x}\|_2^2\right] = \nabla R(\boldsymbol{\theta}). \tag{7}$$

The variance of the stochastic gradient is

$$\mathbb{E}\left[\|G(\boldsymbol{\theta})\|_2^2\right] = \mathbb{E}\left[\left\|\nabla_{\boldsymbol{\theta}}\|f_{\boldsymbol{\theta}}(\mathbf{y}) - \mathbf{x}\|_2^2 - \mathbf{J}_{\boldsymbol{\theta},\mathbf{y}}^T\mathbf{e}\right\|_2^2\right]$$

$$= \mathbb{E}\left[\left\|\nabla_{\boldsymbol{\theta}}\|f_{\boldsymbol{\theta}}(\mathbf{y}) - \mathbf{x}\|_2^2\right\|_2^2\right] + \frac{\sigma_e^2}{n}\|\mathbf{J}_{\boldsymbol{\theta},\mathbf{y}}\|_F^2. \tag{8}$$

We are now ready to prove Theorem 1. Recall that for the theorem, we consider a simple denoising problem where $\mathbf{x} = \mathbf{Uc}$, $\mathbf{c} \sim \mathcal{N}(0, 1/d\mathbf{I})$ and the measurement is $\mathbf{y} = \mathbf{x} + \mathbf{z}$, where $\mathbf{z} \sim \mathbf{N}(0, \sigma_z^2/n\mathbf{I})$. Also, we consider a simple linear estimator $f(\mathbf{y}) = \mathbf{Wy}$ parameterized by the matrix $\mathbf{W} \in \mathbb{R}^{n \times n}$. For this setup, the risk becomes

$$R(\mathbf{W}) = \mathbb{E}\left[\|\mathbf{Wy} - \mathbf{x}\|_2^2\right]$$

$$= \frac{1}{d}\|(\mathbf{W} - \mathbf{I})\mathbf{U}\|_F^2 + \frac{\sigma_z^2}{n}\|\mathbf{W}\|_F^2.$$

Thus, the risk is $m$-strongly convex with $m = \frac{\sigma_z^2}{n}$, since the risk is lower-bounded by the quadratic $\frac{\sigma_z^2}{n}\|\mathbf{W}\|_F^2$. Moreover, the gradient of the risk is

$$\nabla R(\mathbf{W}) = \frac{1}{d}(\mathbf{W} - \mathbf{I})\mathbf{U}\mathbf{U}^T + \frac{\sigma_z^2}{n}\mathbf{W}, \tag{9}$$

and setting this to zero yields the risk-minimizing parameter

$$\mathbf{W}^* = \frac{1}{1 + \sigma_z^2\frac{d}{n}}\mathbf{U}\mathbf{U}^T.$$

Below, we show that the second moment of the risk is $(M, B)$-bounded in that for all $\mathbf{W}$,

$$\mathbb{E}\left[\|G(\mathbf{W})\|_2^2\right] \leq M^2\|\mathbf{W} - \mathbf{W}^*\|_F^2 + B^2, \tag{10}$$

where $\mathbf{W}^*$ is a minimizer of the risk $R(\mathbf{W})$, and where $M = 10/d$ and $B = 12\sigma_z^2\frac{d}{n} + \sigma_e^2(1 + \sigma_z^2)$.

We proceed by applying the following standard result for the convergence of the SGM. For the sake of completeness, the result is proven below.

**Lemma 1.** *Let $f$ be $m$-strongly convex (i.e., $g(\boldsymbol{\theta}) = f(\boldsymbol{\theta}) - \frac{m}{2}\|\boldsymbol{\theta}\|_2^2$ is convex) with minimizer $\boldsymbol{\theta}^*$ and let $G$ be a stochastic gradient that is $(M, B)$-bounded. Assume we start the SGM iterates at $\boldsymbol{\theta}_0 = 0$, with a stepsize $\alpha_k = \frac{2}{m}\frac{1}{2\frac{M^2}{m^2} + k}$. Then*

$$\mathbb{E}\left[\|\boldsymbol{\theta}_k - \boldsymbol{\theta}^*\|_2^2\right] \leq \frac{1}{k-2}\frac{1}{m^2}\left(2M^2 e_0 + B^2\right).$$

Below we show that for all $\mathbf{W}$

$$R(\mathbf{W}) - R(\mathbf{W}^*) \leq (1/d + \sigma_z^2/n)\|\mathbf{W} - \mathbf{W}^*\|_F^2. \tag{11}$$

Thus, an application of Lemma 1 to the RHS yields

$$R(\mathbf{W}_N) - R(\mathbf{W}^*) \leq (1/d + \sigma_z^2/n)\frac{1}{k-2}\left(2\frac{M^2}{m^2}\|\mathbf{W}^*\|_F^2 + \frac{B^2}{m^2}\right)$$

$$\leq \frac{1/d + \sigma_z^2/n}{(\sigma_z^2/n)^2}\frac{1}{k-2}\left(2 + B^2\right),$$

which concludes the proof of Theorem 1.

## B.1 Bound on variance of stochastic gradient, proof of equation (8):

From equation (8), we have

$$\mathbb{E}\left[\|G(\mathbf{W})\|_F^2\right] = \mathbb{E}\left[\left\|\nabla_{\mathbf{W}}\|\mathbf{W}\mathbf{y} - \mathbf{x}\|_2^2\right\|_F^2\right] + \frac{\sigma_e^2}{n}\mathbb{E}\left[\|[\mathbf{x}+\mathbf{z},\ldots,\mathbf{x}+\mathbf{z}]\|_F^2\right]$$

$$= \mathbb{E}\left[\left\|((\mathbf{W}-\mathbf{I})\mathbf{x} + \mathbf{W}\mathbf{z})(\mathbf{x}+\mathbf{z})^T\right\|_F^2\right] + \mathbb{E}\left[\left\|\mathbf{e}(\mathbf{x}+\mathbf{z})^T\right\|_F^2\right]$$

$$= R + \sigma_e^2(1+\sigma_z^2).$$

The term $R$ can be bounded as

$$R = \mathbb{E}\left[\left\|((\mathbf{W}-\mathbf{W}^*+\mathbf{W}^*-\mathbf{I})\mathbf{x} + (\mathbf{W}-\mathbf{W}^*+\mathbf{W}^*)\mathbf{z})(\mathbf{x}+\mathbf{z})^T\right\|_F^2\right]$$

$$= \mathbb{E}\left[\left\|(\mathbf{W}-\mathbf{W}^*)(\mathbf{x}\mathbf{x}^T+\mathbf{x}\mathbf{z}^T+\mathbf{z}\mathbf{x}^T+\mathbf{z}\mathbf{z}^T) + (\mathbf{W}^*-\mathbf{I})(\mathbf{x}\mathbf{x}^T+\mathbf{x}\mathbf{z}^T) + \mathbf{W}^*(\mathbf{z}\mathbf{x}^T+\mathbf{z}\mathbf{z}^T)\right\|_F^2\right]$$

$$\leq 2\mathbb{E}\left[\left\|(\mathbf{W}-\mathbf{W}^*)(\mathbf{x}\mathbf{x}^T+\mathbf{x}\mathbf{z}^T+\mathbf{z}\mathbf{x}^T+\mathbf{z}\mathbf{z}^T)\right\|_F^2\right]$$

$$+ 2\mathbb{E}\left[\left\|(\mathbf{W}^*-\mathbf{I})(\mathbf{x}\mathbf{x}^T+\mathbf{x}\mathbf{z}^T) + \mathbf{W}^*(\mathbf{z}\mathbf{x}^T+\mathbf{z}\mathbf{z}^T)\right\|_F^2\right]$$

$$= 2R_1 + 2R_2.$$

We next bound $R_1$ and $R_2$.

**Bound on $R_1$:** To bound $R_1$, first note that for $\mathbf{x} \sim \mathcal{N}(0, 1/d\mathbf{I})$,

$$\mathbb{E}\left[\left\|\mathbf{A}\mathbf{x}\mathbf{x}^T\right\|_F^2\right] = \operatorname{trace}\mathbb{E}\left[\mathbf{A}\mathbf{x}\mathbf{x}^T\mathbf{x}\mathbf{x}^T\mathbf{A}^T\right] \leq 2\operatorname{trace}\mathbb{E}\left[\mathbf{A}\mathbf{x}\mathbf{x}^T\mathbf{A}^T\right] = 2\mathbb{E}\left[\|\mathbf{A}\mathbf{x}\|_2^2\right] = \frac{2}{d}\|\mathbf{A}\mathbf{U}\|_F^2.$$

Similarly, for $\mathbf{x} \sim \mathcal{N}(0, 1/d\mathbf{I})$ and $\mathbf{z} \sim \mathcal{N}(0, \sigma_z^2/n\mathbf{I})$

$$\mathbb{E}\left[\left\|\mathbf{A}\mathbf{x}\mathbf{z}^T\right\|_F^2\right] = \operatorname{trace}\mathbb{E}\left[\mathbf{A}\mathbf{x}\mathbf{z}^T\mathbf{z}\mathbf{x}^T\mathbf{A}^T\right] = \sigma_z^2\operatorname{trace}\mathbb{E}\left[\mathbf{A}\mathbf{x}\mathbf{x}^T\mathbf{A}^T\right] = \sigma_z^2\mathbb{E}\left[\|\mathbf{A}\mathbf{x}\|_2^2\right] = \frac{\sigma_z^2}{d}\|\mathbf{A}\mathbf{U}\|_F^2,$$

and

$$\mathbb{E}\left[\left\|\mathbf{A}\mathbf{z}\mathbf{x}^T\right\|_F^2\right] = \operatorname{trace}\mathbb{E}\left[\mathbf{A}\mathbf{z}\mathbf{x}^T\mathbf{x}\mathbf{z}^T\mathbf{A}^T\right] = \operatorname{trace}\mathbb{E}\left[\mathbf{A}\mathbf{z}\mathbf{z}^T\mathbf{A}^T\right] = \mathbb{E}\left[\|\mathbf{A}\mathbf{z}\|_2^2\right] = \frac{\sigma_z^2}{n}\|\mathbf{A}\|_F^2,$$

Also similarly,

$$\mathbb{E}\left[\left\|\mathbf{A}\mathbf{z}\mathbf{z}^T\right\|_F^2\right] \leq \frac{2\sigma_z^4}{n}\|\mathbf{A}\|_F^2.$$

It follows that

$$R_1 \leq \|\mathbf{W}-\mathbf{W}^*\|_F^2\left(\frac{2}{d} + \frac{2\sigma_z^2}{d} + \frac{2\sigma_z^4}{n}\right).$$

**Bound on $R_2$:** Note that

$$R_2 \leq \|(\mathbf{W}^*-\mathbf{I})\mathbf{U}\|_F^2\frac{2+\sigma_z^2}{d} + \|\mathbf{W}^*\|_F^2\frac{\sigma_z^2+2\sigma_z^4}{n}$$

$$\leq \left(\sigma_z^2\frac{d}{n}\right)^2 d\frac{2+\sigma_z^2}{d} + d\frac{\sigma_z^2+2\sigma_z^4}{n}$$

$$\leq 3\left(\sigma_z^2\frac{d}{n}\right)^2 + d\frac{\sigma_z^2+2\sigma_z^4}{n}$$

$$\leq 6\frac{d}{n}\sigma_z^2.$$

where we used that $\|(\mathbf{W}^*-\mathbf{I})\mathbf{U}\|_F^2 \leq \left(\sigma_z^2\frac{d}{n}\right)^2 d$ and $\|\mathbf{W}^*\|_F^2 \leq d$, and for the last inequality we used that $\sigma_z^2 \leq 1$, by assumption.

Application of the bounds on $R_1$ and $R_2$ above concludes the proof of Equation (8).

## B.2 Proof of equation (11)

Note that the risk can be written as
$$R(\mathbf{W}) = \mathbb{E}\left[\|\mathbf{W}\mathbf{y} - \mathbf{x}\|_2^2\right] = \mathbb{E}\left[\|\mathbf{Y}\mathbf{w} - \mathbf{x}\|_2^2\right], \tag{12}$$
where $\mathbf{w} = [\mathbf{w}_1^T, \ldots, \mathbf{w}_n^T]^T \in \mathbb{R}^{n^2}$ and $\mathbf{Y} \in \mathbb{R}^{n \times n^2}$ is a block diagonal matrix containing the row-vector $\mathbf{y}^T$ on the diagonal. With this notation,

$$
\begin{aligned}
R(\mathbf{W}) &= \mathbb{E}\left[\|\mathbf{Y}\mathbf{w} - \mathbf{x}\|_2^2\right] \\
&= \mathbb{E}\left[\|\mathbf{Y}\mathbf{w} - \mathbf{Y}\mathbf{w}^* + \mathbf{Y}\mathbf{w}^* - \mathbf{x}\|_2^2\right] \\
&= \mathbb{E}\left[\|\mathbf{Y}(\mathbf{w} - \mathbf{w}^*)\|_2^2\right] + 2\mathbb{E}\left[(\mathbf{Y}(\mathbf{w} - \mathbf{w}^*)^T(\mathbf{Y}\mathbf{w}^* - \mathbf{x})\right] + \mathbb{E}\left[\|\mathbf{Y}\mathbf{w}^* - \mathbf{x}^*\|_2^2\right] \\
&= \mathbb{E}\left[\|\mathbf{Y}(\mathbf{w} - \mathbf{w}^*)\|_2^2\right] + 2(\mathbf{w} - \mathbf{w}^*)^T\mathbb{E}\left[\mathbf{Y}^T(\mathbf{Y}\mathbf{w}^* - \mathbf{x})\right] + R(\mathbf{W}^*) \\
&\overset{\mathrm{i}}{=} \mathbb{E}\left[\|\mathbf{Y}(\mathbf{w} - \mathbf{w}^*)\|_2^2\right] + R(\mathbf{W}^*) \\
&\leq (1/d + \sigma_z^2/n)\|\mathbf{W} - \mathbf{W}^*\|_F^2 + R(\mathbf{W}^*)
\end{aligned}
$$
where equality i follows from $0 = \nabla R(\mathbf{w}^*) = \mathbb{E}\left[2\mathbf{Y}^T(\mathbf{Y}\mathbf{w}^* - \mathbf{x})\right]$ for a minimizer $\mathbf{w}^*$, and the last inequality follows from $\mathbf{y} = \mathbf{U}\mathbf{c} + \mathbf{z}$ where $\mathbf{c} \sim \mathcal{N}(0, 1/d\mathbf{I})$ and $\mathbf{x} \sim \mathcal{N}(0, \sigma_z^2/2\mathbf{I})$.

## B.3 Proof of Lemma 1

We start by upper bounding the expectation $e_k = \mathbb{E}\left[\|\boldsymbol{\theta}_k - \boldsymbol{\theta}^*\|_2^2\right]$. We write $G_k(\boldsymbol{\theta}_k) = G(\boldsymbol{\theta}_k)$ for the stochastic gradient at iteration $k$ to emphasize that at each iteration, the gradient is drawn independently from the gradient at other iterations. First note that

$$
\begin{aligned}
\|\boldsymbol{\theta}_{k+1} - \boldsymbol{\theta}^*\|_2^2 &= \|\boldsymbol{\theta}_k - \alpha_k G_k(\boldsymbol{\theta}_k) - \boldsymbol{\theta}^*\|_2^2 \\
&= \|\boldsymbol{\theta}_k - \boldsymbol{\theta}^*\|_2^2 - 2\alpha_k \langle G_k(\boldsymbol{\theta}_k), \boldsymbol{\theta}_k - \boldsymbol{\theta}^*\rangle + \alpha_k^2\|G_k(\boldsymbol{\theta}_k)\|_2^2.
\end{aligned}
$$
The expectation of the middle term above is
$$\mathbb{E}\left[\langle G_k(\boldsymbol{\theta}_k), \boldsymbol{\theta}_k - \boldsymbol{\theta}^*\rangle\right] = \mathbb{E}\left[\langle \mathbb{E}\left[G_k(\boldsymbol{\theta}_k)|\boldsymbol{\theta}_k\right], \boldsymbol{\theta}_k - \boldsymbol{\theta}^*\rangle\right] = \mathbb{E}\left[\langle \nabla h(\boldsymbol{\theta}_k), \boldsymbol{\theta}_k - \boldsymbol{\theta}^*\rangle\right], \tag{13}$$
where we used that the random variable $G_k$ is independent of the random variable $G_i, i < k$, since each stochastic gradient is drawn independently, and is therefore independent of the iterate $\boldsymbol{\theta}_k$. Thus, iterating the expectation allows us to replace the stochastic gradient by the gradient.

Using the assumption that the stochastic gradient $G$ is $(M, B)$-bounded yields
$$e_{k+1} \leq (1 + \alpha_k^2 M^2)e_k - 2\alpha_k\mathbb{E}\left[\langle \nabla \mathbf{f}(\boldsymbol{\theta}_k), \boldsymbol{\theta}_k - \boldsymbol{\theta}^*\rangle\right] + \alpha_k^2 B^2. \tag{14}$$
With $\nabla h(\boldsymbol{\theta}^*) = 0$, we have that
$$\langle \nabla h(\boldsymbol{\theta}_k), \boldsymbol{\theta}_k - \boldsymbol{\theta}^*\rangle = \langle \nabla h(\boldsymbol{\theta}_k) - \nabla h(\boldsymbol{\theta}^*), \boldsymbol{\theta}_k - \boldsymbol{\theta}^*\rangle \geq m\|\boldsymbol{\theta}_k - \boldsymbol{\theta}^*\|_2,$$
where the inequality follows from $h$ being $m$-strongly convex. Application of this inequality to (14) yields
$$e_{k+1} \leq (1 + \alpha_k^2 M^2 - 2\alpha_k m)e_k + \alpha_k^2 B^2. \tag{15}$$
Since we choose the stepsize as $\alpha_k = \frac{2}{m}\frac{1}{2\beta + k}$ with $\beta = \frac{M^2}{m^2}$ we have $\alpha_k \leq \frac{m}{M^2}$. Then Equation (15) yields
$$e_{k+1} \leq (1 - \alpha_k m)e_k + \alpha_k^2 B^2.$$
Unrolling the iterations gives

$$
\begin{aligned}
e_k &\leq (1 - \alpha_{k-1}m)e_{k-1} + \alpha_{k-1}^2 B^2 \\
&\leq (1 - \alpha_{k-1}m)(1 - \alpha_{k-2}m)e_{k-2} + (1 - \alpha_{k-1}m)\alpha_{k-2}^2 B^2 + \alpha_{k-1}^2 B^2 \\
&\leq \prod_{i=0}^{k-1}(1 - \alpha_i m)e_0 + B^2 \sum_{i=0}^{k-1}\alpha_i^2 \prod_{j=i+1}^{k-1}(1 - \alpha_j m) \\
&= \prod_{i=0}^{k-1}\left(1 - \frac{2}{2\beta + i}\right)e_0 + B^2 \sum_{i=0}^{k-1}\left(\frac{1}{m}\frac{2}{2\beta + i}\right)^2 \prod_{j=i+1}^{k-1}\left(1 - \frac{2}{2\beta + j}\right).
\end{aligned}
$$

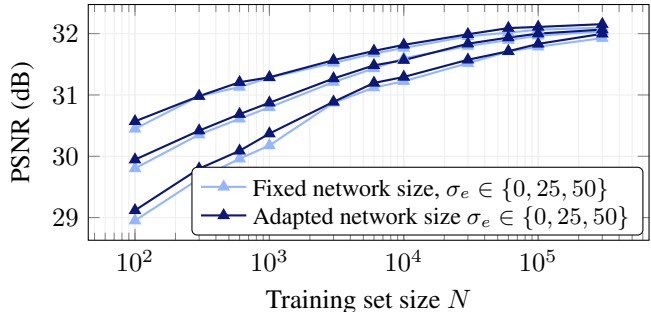

Figure 7: **Effect of adapting the network size for denoising.** For each training set size the dark blue curves show the best performing network over a range of network sizes (see Figure 8 for all results). Light blue curves show the performance for a fixed network size. Since the performance curves of supervised training ($\sigma_e = 0$, top curves) and of self-supervised training ($\sigma_e \in \{25, 50\}$, middle and bottom curves) are affected very similarly from fixing the network size, the relative comparison between supervised and self-supervised training remain valid.

Using that $\prod_{j=i}^{k} \left(1 - \frac{2}{j}\right)^2 = \frac{(i-2)(i-1)}{(k-1)k}$, we get

$$
\begin{aligned}
e_k &\leq \frac{(2\beta - 2)(2\beta - 1)}{(k-2)(k-1)} e_0 + \frac{B^2}{m^2} \sum_{i=0}^{k-1} \left(\frac{2}{2\beta + i}\right)^2 \frac{(2\beta + i - 1)(2\beta + i)}{(k-2)(k-1)} \\
&\leq \frac{4\beta^2}{(k-2)^2} e_0 + \frac{B^2}{m^2} \frac{1}{k-2} \\
&\leq \frac{1}{k-2} \left(2\beta e_0 + \frac{B^2}{m^2}\right),
\end{aligned}
$$

which concludes the proof.

## C   Additional material for Section 4: Self-supervised image denoising

### C.1   Gaussian image denoising

The Gaussian image denoising experiments are conducted on the following datasets. From the ImageNet dataset [37] we reserve 80/230 images for validation/testing, and create training sets $\mathcal{S}_N$ of size $N$ ranging from 100 to 300k images and $\mathcal{S}_i \subset \mathcal{S}_j$ for $i < j$. For the training set we center crop images to size $128 \times 128$.

Further, all following experiments that rely on the U-net as the network architecture, use a U-net with two blocks in the encoder and decoder part respectively and skip connections between blocks. Each block consists of two convolutional layers with LeakyReLU activation and instance normalization [41] after every layer, where the number of channels is doubled (halved) after every block in the encoder (decoder). The downsampling in the encoder is implemented as average pooling and the upsampling in the decoder as transposed convolutions. As proposed by Zhang et al. [48] we train a residual denoiser that learns to predict $\mathbf{y} - \mathbf{y}'$ instead of directly predicting $\mathbf{y}'$, which improves performance.

All experiments were conducted on NVIDIA A40, NVIDIA RTX A6000 and NVIDIA Quadro RTX 6000 GPUs. We measure the time in GPU hours until the best epoch according to the validation loss resulting in about 700 GPU hours for the experiments on self-supervised Gaussian denoising presented in Figure 2.

### C.1.1   Noise2noise self-supervised denoising with a U-net of varying size

The results in Figure 2 are for a U-net of *fixed network size*, which is the natural setup since we're interested in studying differences of training schemes, and therefore we have to keep the network architecture constant.

However, for different training set sizes, networks of different sizes perform best; if the training set size is larger, typically a larger network performs best. In the following, we therefore perform a comparison of self-supervised training and supervised training, where we *adapt the network size* along with the training set size, $N$.

We find that adapting the network size to the training set size improves the performance of models trained in a supervised and a self-supervised fashion similarly, thus our findings on self-supervised versus supervised training continue to hold even if we adopt the network size as we change the training set size.

To show this, we follow an approach from the scaling law literature, where the goal is to observe the isolated effect of one resource on the model performance by providing unlimited access to all other resources. In the context of image reconstruction Klug and Heckel [22], investigate the performance as a function of the training set size without limiting network size and invested compute, i.e., by only considering the best performing model per training set size over a wide range of network sizes and by training each model until the performance converges on the validation set. Similar work has been done in other fields including computer vision [47] and natural language processing [20].

Figure 8 shows our additional results for adapting the network size for noise2noise self-supervised training with target noise levels $\sigma_e \in \{0, 25, 50\}$. The panels on the right show the performance as a function of the network size for a fixed training set size. The left panels show the performance as a function of the training set size, where only the best model per training set size is considered.

Figure 7 compares the three curves with adapted network size from the left panels in Figure 8 to the performance curves for a fixed network size from Figure 2.

We chose the fixed network to have a moderate size (7.4M) compared to the smallest (0.1M) and largest networks (46.5M) considered for adapting the network size. As a consequence in Figure 7 the smallest/largest training set sizes gain most performance from adapting to a smaller/larger network size. However, the gain in performance is small, which is also indicated by the relatively flat curves in the right panels of Figure 8, which show the scaling of the reconstruction performance as a function of the network size. Further, the three training methods, supervised and self-supervised training with $\sigma_e \in \{25, 50\}$, are affected similarly from fixing the network size. Hence, our findings regarding a relative comparison of supervised and self-supervised methods are not affected.

**Training details for varying the network size.** The left column in Figure 8 is a summary of all models that we trained for Gaussian denoising with a U-net under different levels of noise on the training targets $\sigma_e \in \{0, 25, 50\}$. For different training set sizes $N \in \{0.1, 0.3, 0.6, 1, 3, 6, 10, 30, 60, 100, 300\}$ thousand images we varied the number of channels in the first layer of the U-net in $\{16, 32, 64, 128, 192, 256, 320\}$ corresponding to $P = \{0.1, 0.5, 1.9, 7.4, 16.7, 30.0, 46.5\}$ million network parameters.

For small training set sizes the curves in Figure 8 use the best performance out of up to 5 training runs with different random network initialization, indicated by the additional points underneath the different curves. In Figure 7 only the best runs are reported for brevity. As the variation among runs decreases as the training set size increases, a single run is performed for large training set sizes, and further runs are only added in case of obvious outliers.

We tested two different batch sizes 1 and 10 for training set sizes up to 60k images and found that while models trained with small training set sizes significantly benefit from batch size 1, models trained with larger training set sizes are indifferent to the choice of the batch size. Hence, we set the batch size to 1 for training set sizes $N \leq 6k$ and to 10 otherwise.

All models are trained with the Adam optimizer [21] with $\beta_1 = 0.9$ and $\beta_2 = 0.999$. As we train on different combinations of network size, batch size and training set size we have to adjust the initial learning rate accordingly.

We found that the following automated scheme for learning rate annealing reliably finds an initial learning rate that lies within the range of suitable learning rates. Starting from a small learning rate $1.25 \times 10^{-6}$ we double the learning rate after every epoch as long as the validation error improves. If training with a learning rate for 3 epochs does not yield an improvement, the annealing is terminated. We then continue the training with the model checkpoint corresponding to a learning rate 4-times smaller than the current learning rate.

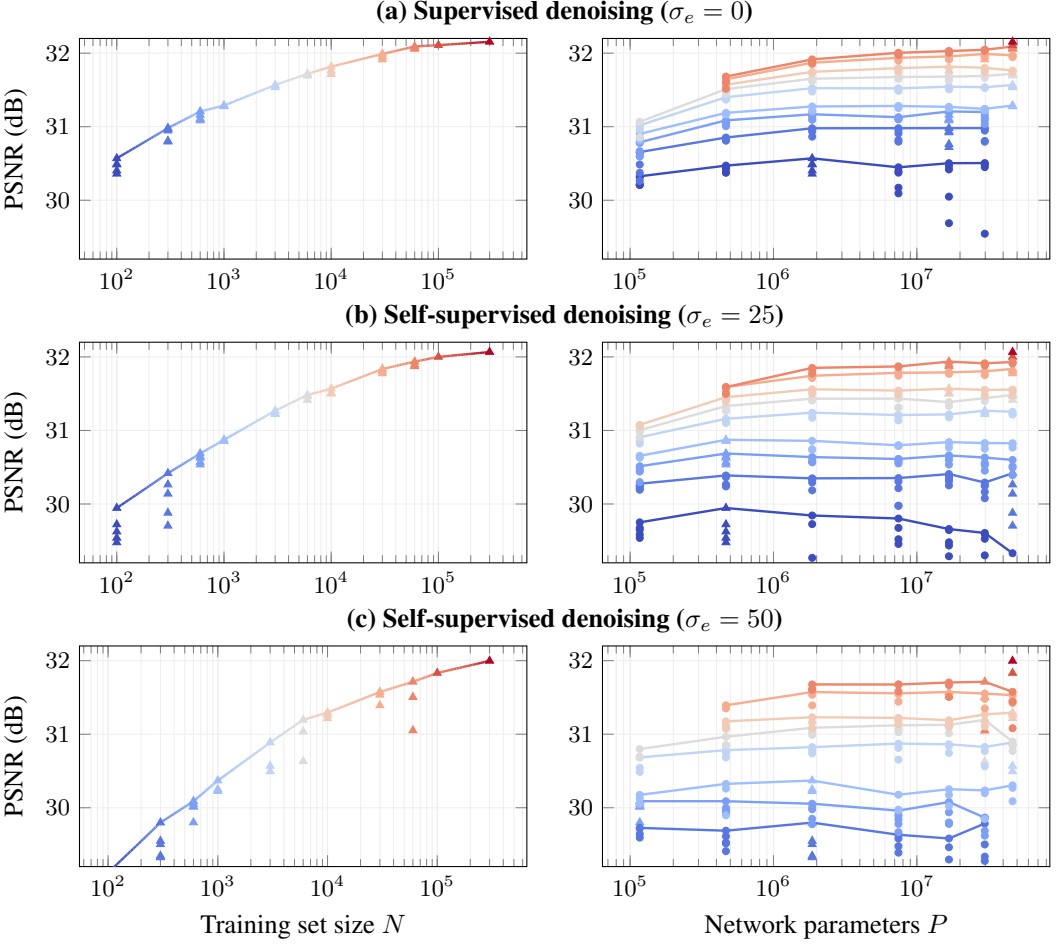

Figure 8: **Complete results for adapting the network size for denoising.** Each curve on the right is obtained by fixing the training set size and sweeping across different network sizes. Then, the curve on the left is created by taking the best performing network for each training set size. Colors in the plots on the left and right correspond to the same training set size.

Once the learning rate is set, we apply an automated learning rate decay. We halve the learning rate once the validation PSNR did not improve for eight consecutive epochs. If the learning rate was decayed twice in a row without an improvement, we terminate the training. Then, the model with the best validation PSNR is picked for evaluation on the test set. Picking the model based on the validation error was necessary since we faced overfitting during training for $\sigma_e \in \{25, 50\}$, as shown in the example in Figure 9. The overfitting is to be expected as we fix the noise per training example over all epochs on both noisy inputs $\mathbf{y}$ and noisy targets $\mathbf{y}'$.

To perform early stopping we compute a self-supervised validation PSNR between measurements $\mathbf{y}$ and $\mathbf{y}'$ of the images $\mathbf{x}$ in the validation set. It is well known that the self-supervised validation PSNR is proportional to the actual PSNR, see [9] for a discussion. As the self-supervised training error approximates the supervised one, so does the self-supervised validation error. Figure 9 shows an example for the correlation between self-supervised and supervised validation error.

### C.1.2 Noise2noise self-supervised denoising with a U-net of fixed size

The results in Figure 2 for noise2noise self-supervised denoising were obtained with a fixed network size with 128 channels in the first layer resulting in 7.4M parameters. The training procedure for fixed size networks is similar to training with networks of varying size described in Appendix C.1.1.

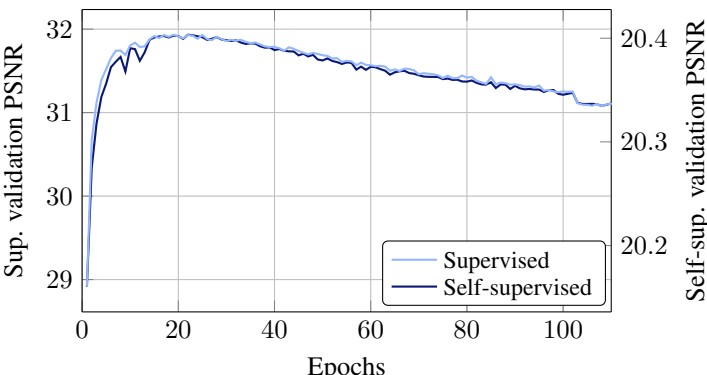

Figure 9: **Overfitting and supervised vs. self-supervised validation PSNR for denoising.** In this example, supervised and self-supervised validation PSNR of a network trained in a noise2noise self-supervised manner (with $\sigma_e = 25$) improves until epoch 20. At epoch 20, performance declines due to overfitting to the fixed noise realizations on the training inputs and targets. As supervised and self-supervised PSNR behave qualitatively the same we can use the self-supervised PSNR to pick the best model over all epochs. This experiment uses a training set of size 100k and a U-net with 320 channels in the first layer.

For $N \leq 6000$ images, we reused the networks trained for the adaptive network scenario from Figure 8 since we already have 128-channeled networks for them. The training procedure for those networks is discussed in Appendix C.1.1.

For $N \geq 10.000$ images, we used a batch size of 1 and as the network size is fix, we also fix the initial learning rate to $6.4 \times 10^{-4}$ and then applied learning rate decay as described in Appendix C.1.1. For the largest training set size $N = 300k$ we trained once with initial learning rate $6.4 \times 10^{-4}$ and once with $1.6 \times 10^{-4}$ and picked the run that performed better.

As in Appendix C.1.1 we picked the best out of up to three training runs from different random initializations to account for the variance between runs for small and medium sized training sets.

### C.1.3 Noisier2noise self-supervised denoising with a U-net

The performance curve for noisier2noise self-supervised denoising in Figure 2 is obtained with a U-net of fixed size with 128 channels in the first layer and 7.4M network parameters in order to match the setup for noise2noise in Appendix C.1.2.

In noisier2noise, we have only one noisy measurement per clean image, which is used as the training target. In the simplest version presented by Moran et al. [30] the training input is created by injecting additional noise of the same distribution and variance to the noisy image. That means we add Gaussian noise with noise level $\sigma = 25$ to the images in the training set and train the network to predict the original noisy training image from its noisier version. As we have control over the noise that we inject during training, it is re-sampled in every epoch to improve performance.

Similar to training with the noise2noise loss with fixed network size described in Appendix C.1.2 we use a batch size of 1, the Adam optimizer with $\beta_1 = 0.9$ and $\beta_2 = 0.999$ and an initial learning rate of $6.4 \times 10^{-4}$. To be consistent with Appendix C.1.2 we also tried an initial learning of $1.6 \times 10^{-4}$ for $N = 300k$, which performed slightly better.

We apply the same scheme for learning rate decay as described in Appendix C.1.2, where the self-supervised validation loss is computed between the noisy training target and the network output. We did not observe overfitting in the noisier2noise setup since the additional noise on the noisier training inputs is re-sampled in every epoch.

As in Appendix C.1.2 we picked the best out of up to three training runs from different random initializations to account for the variance between runs for small and medium sized training sets.

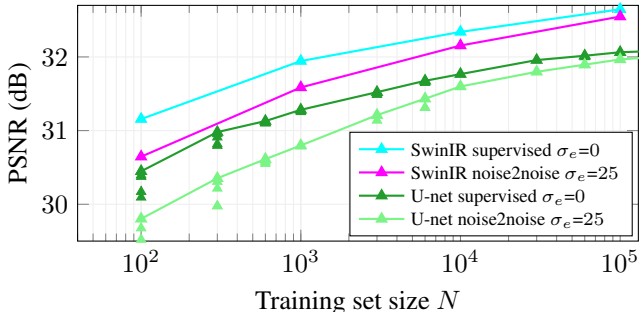

Figure 10: **Gaussian image denoising with a SwinIR vs. U-net**. While the SwinIR outperforms the U-net in terms of absolute performance, the performance of both models trained in a noise2noise manner approaches the performance when trained in a supervised manner as the number of training images increases.

### C.1.4 Neighbor2neighbor self-supervised denoising with a U-net

To obtain the results for neighbor2neighbor in Figure 2 we trained the same U-net with 128 channels in its first layer, corresponding to 7.4M parameters as in the experiments for noisier2noise and noise2noise. As before, we set the batch size to 1 and use the Adam optimizer with $\beta_1 = 0.9$ and $\beta_2 = 0.999$.

In neighbor2neighbor training the given noisy example is subsampled into two images that are used as training input and target. We follow the approach described in Huang et al. [18] and we refer to their description for more details on the subsampling, the loss function and the regularization used during training.

As proposed by Huang et al. [18] we fix the number of training epochs to 100 for all training set sizes and halved the learning rate once in every 20 epochs. However, with the total number of epochs fixed the convergence of the network as the training set size changes might depend on the choice of the initial learning rate. Hence, instead of fixing the initial learning rate as for noise2noise and noisier2noise in Appendix C.1.3 and C.1.2, we ablate for a good choice of initial learning rate in the range of $10^{-5}$ to $10^{-2}$ for every training set size. Contrary to noise2noise and noisier2noise, for the neighbor2neighbor loss computing a self-supervised validation loss based only on a noisy validation set did not correlate well with the supervised validation loss. Hence, we assumed access to a version of our validation set that contains ground truth images to evaluate the results of the learning rate ablation study and to pick the best performing model over the training epochs. The curve in Figure 2 contains only the performance of the run with the best learning rate per training set size.

Same as in our noise2noise and noisier2noise experiments we picked the best out of up to three training runs from different random initializations to account for the variance between runs for small and medium sized training sets.

### C.1.5 Noise2noise self-supervised denoising with a SwinIR

In Section 4 Figure 2 we empirically determined the sample complexity of noise2noise self-supervised and supervised Gaussian denoising with a U-net and showed that as the training set size gets large a model trained in a self-supervised manner approaches the performance of a model trained in a supervised manner. However, we expect that our results for comparing supervised and self-supervised training schemes translate qualitatively to other network architectures.

In this section we confirm this by re-running our experiments with the SwinIR [26] architecture, a recent state-of-the-art model for image denoising based on the Swin Transformer [27].

Figure 10 shows the performance of training a SwinIR on the training sets of size $N \in \{0.1, 1, 10, 100\}$ thousand image patches with a supervised and a noise2noise self-supervised loss with noise level on the training targets $\sigma_e = 25$. As expected, the SwinIR outperforms the U-net (curves are taken from Figure 2) on all training set sizes. However the relative performance gap between the models trained in a supervised and self-supervised way is similar and at 100k training examples the gap is reduced to 0.098/0.097dB for the SwinIR/U-net.

We train the default SwinIR for color image denoising from Liang et al. [26] with 11.5M network parameters. The networks are trained with the MSE loss, the Adam optimizer and learning rates $\{2e-4, 1e-4, 8e-5, 8e-5\}$ for training set sizes $N \in \{0.1, 1, 10, 100\}$ thousand training examples respectively. Depending on the training set size, the networks trained with supervised/self-supervised loss are trained for $\{100, 87, 78, 40\}/\{100, 87, 50, 26\}$ epochs, where we need less epochs for self-supervised training due to overfitting similar to what was observed for noise2noise training with the U-net in Figure 9. The initial learning rate is decayed by a factor of 2 at 60%, 75% and 90% of the total number of training epochs.

## C.2  Real-world camera noise denoising

In this section we provide additional details for the experiments on real-world camera noise denoising with the SID [1] dataset presented in Figure 3.

We train the same U-net as described in Appendix C.1, with 56 channels in the first layer resulting in 1.4M network parameters.

Each training set consists of non-overlapping patches of size $128 \times 128$ drawn randomly from all available patches. From the 160 scene instances in the medium SIDD training set we use all scenes except scenes 0199 and 0200 as the remaining scenes already constitute to 125k patches and the largest training set we consider contains only 100k patches.

For self-supervised training input and target are cropped from the two noisy images given per scene in the medium SID dataset. For supervised training the noisy target is replaced with the ground truth image estimated from all noisy images of this scene as described in Abdelhamed et al. [1]. For the different runs with a fixed training set size displayed in Figure 3, training patches and network initializations are drawn independently between runs.

Training and testing is performed in the raw-RGB space, i.e., on single-channel images where every square of $2 \times 2$ pixels contains one R,G,G and B value. The order of the 4 values varies over the dataset depending on the Bayer pattern of the various cameras used. Given the Bayer pattern of an image we rearrange the squares of size $2 \times 2$ pixels to follow the order RGGB. Finally the raw images are packed into a 4-channel representation corresponding to their RGGB values before processed by the network.

All networks are trained with the Adam optimizer with an initial learning rate of 0.00032, which is halved at a fixed set of epochs depending on the training set size.

# D  Additional material for Section 5: Self-supervised compressive sensing

## D.1  Compressive sensing for natural images

For the experiments in Section 5.1 we train the same U-net as described in Appendix C.1, but with 3 blocks in the encoder/decoder and with 24 channels in the first layer resulting in about 1M network parameters, and unlike in Appendix C.1 the network directly predicts the target measurement $\mathbf{y}'$ instead of the residual $\mathbf{y} - \mathbf{y}'$ during training. The input to the network consists of a complex-valued coarse reconstruction $\mathbf{F}^{-1}\mathbf{y}$ in the image domain, which we normalize to zero-mean and standard deviation one before handed to the U-net. Further, the real and complex part are forwarded to the U-net as two separate input channels. The output of the network is denormalized before transformed to the frequency domain in which the (masked in case of self-supervised training) $\ell_2$ training loss is computed. The test performance is evaluated between the absolute value of the network output and the ground truth image.

The networks are trained using the Adam optimizer [21] with an initial learning rate of $10^{-3}$. The networks are evaluated on the validation set in every second epoch and the learning rate is decayed by a factor of 10 if there is no improvement for certain number of consecutive validations. This patience parameter is gradually decreased from 20 for the smallest to 6 for the largest training set size. The training is terminated after training for a few epochs with the minimal learning rate $10^{-5}$. Only the best model according to the validation loss is evaluated on the test set, which we then report in Figure 5.The validation/test set consists of 80/300 patches. The batch size is set to 1.

All experiments were conducted on a NVIDIA A40 GPU. We measure the time in GPU hours until the best epoch according to the validation loss resulting in about 500 GPU hours for the experiments in Figure 5.

## D.2 Compressive sensing accelerated MRI

The data from the multi-coil fastMRI brain dataset consists of images of different contrasts. In our experiments we use the subset of about 55k AXT2-weighted images corresponding to a single contrast. We use 50k of those images to design training sets $\mathcal{S}_N$ of varying size $N$ with $\mathcal{S}_i \subset \mathcal{S}_j$ for $i < j$, 300 for validation and the remaining 4700 for testing. The images for validation and testing are taken from different subjects than for training.

The test performance for a given measurement is evaluated between the magnitude of the complex valued network reconstruction and the ground truth image. Note that the estimated sensitivity maps are strictly zero outside the region in which the object that we aim to reconstruct is located. Since we compute the training loss in the k-space after applying the sensitivity maps to the network output, the network output does not receive any supervision in regions, where the sensitivity maps are zero. Hence, in order to compute meaningful test scores on the magnitude of the network output we apply binary masking to remove artifacts in the background of the reconstructed images. The estimated sensitivity maps are normalized in a way such that a binary mask $\mathbf{B} \in \mathbb{R}^{n \times n}$ can be obtained as $\mathbf{B} = \sum_{j=1}^{C} \mathbf{S}_j^* \mathbf{S}_j$.

### D.2.1 Compressive sensing accelerated MRI with a U-net

For the results in Figure 6 we train the same U-net as described in Appendix D.1, but with 4 blocks in the encoder/decoder and with 64 channels in the first layer resulting in about 31M network parameters. We use the same input/output normalization as in Appendix D.1.

The networks are trained with the RMSprop optimizer as it is the default in the fastMRI repository [46] and with an initial learning rate of $10^{-3}$. The networks are evaluated on the validation set after every epoch and the learning rate is decayed by a factor of 10 if no improvement is observed for 10 consecutive epochs. The training is terminated after training 10 epochs with the minimal learning rate of $10^{-6}$. The best model according to the validation loss is evaluated on the test set.

All experiments were conducted on a NVIDIA A40 GPU. We measure the time in GPU hours until the best epoch according to the validation loss resulting in about 600 GPU hours for the experiments presented in Figure 6.

### D.2.2 Compressive sensing accelerated MRI with a VarNet

In this section we provide additional results on supervised versus self-supervised compressive sensing MRI, where we replace the U-net used in Figure 6 with the state-of-the-art end-to-end VarNet [38]. As our analysis pertains to comparing different training schemes, we expect our results to translate qualitatively to other types of network architectures.

As Figure 11 shows the VarNet outperforms the U-net significantly. However, similar to the U-net the relative performance gap between models trained in a noise2noise self-supervised way and trained in a supervised way is small already for small training set sizes.

For training the VarNets we follow the implementation from the official fastMRI repository [46][1], where the VarNet consists of a cascade of U-nets with intermediate data consistency steps and an additional U-net that estimates the sensitivity maps used during the data consistency steps and to compute the input to the first U-net.

We use a VarNet with 6 cascades, where each U-net consists of 4 blocks in the encoder and decoder and 14 channels in the top layer resulting in a total of about 9.4M network parameters. The VarNet outputs an estimate $\hat{\mathbf{y}}_{j,\text{full}}$ of the fully sampled k-space for $j = 1, \ldots, C$ coils. We us the pre-computed sensitivity maps $\mathbf{S}_j$ estimated with ESPIRiT [39] instead of the sensitivity maps estimated

---

[1] https://github.com/facebookresearch/fastMRI

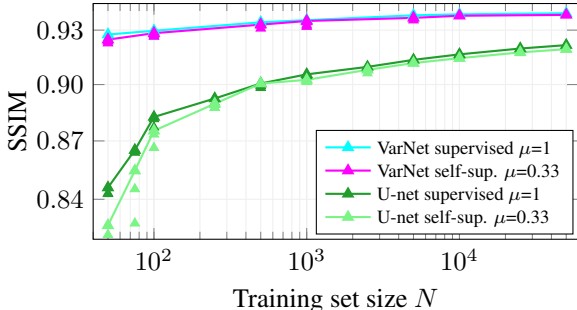

Figure 11: **Compressive sensing MRI with a VarNet vs. U-net.** While the VarNet outperforms the U-net in absolute numbers, for both types of network architecture the performance gap between supervised and self-supervised trained models vanishes already at small training set sizes.

by the network to compute the final reconstructed image as

$$\hat{\mathbf{x}} = \left| \sum_{j=1}^{C} \mathbf{S}_j^* \mathbf{F}^{-1} \hat{\mathbf{y}}_{j,\text{full}} \right|.$$

This is analogous to how we compute the reference ground truth images; See Section 5.2.

Depending on the training set size $N \in \{50, 100, 500, 1000, 5000, 10000, 50000\}$ the model is trained for $N_{\text{ep}} \in \{150, 130, 125, 120, 90, 75, 55\}$ epochs. All networks are trained with the RMSprop optimizer with an initial learning rate of $10^{-3}$, which is decayed by a factor of 10 once for the last 15 and again for the last 5 training epochs.

# E   Details for computing estimates of the variance of the stochastic gradient

In this section we provide additional details on the histograms of the normalized variance of the stochastic gradients, i.e., normalized estimates of the MSE $\|\nabla \ell_{\text{SS}}(f_{\boldsymbol{\theta}}(\mathbf{y}_i), \mathbf{y}_i') - \nabla R(\boldsymbol{\theta})\|_2^2$ after one epoch of training presented in Figures 2, 3, 5 and 6.

We consider the gradients with respect to the network weights $\boldsymbol{\theta}$ after training one epoch with the supervised loss. Note that the gradient distribution changes at each epoch.

We compute stochastic gradients $\nabla \ell_{\text{SS}}(f_{\boldsymbol{\theta}}(\mathbf{y}_i), \mathbf{y}_i')$ for $i = 1, \ldots, N$ with $N = 10000$ images. The gradient of the risk $\nabla R(\boldsymbol{\theta})$ is estimated with the empirical gradient of the supervised loss, i.e.,

$$\nabla \hat{R}(\boldsymbol{\theta}) = \frac{1}{N} \sum_{i=1}^{N} \nabla \ell(f_{\boldsymbol{\theta}}(\mathbf{y}_i), \mathbf{x}_i).$$

Finally, the normalized variance of one the $i$-th stochastic gradient is computed as

$$\frac{\left\| \nabla \ell_{\text{SS}}(f_{\boldsymbol{\theta}}(\mathbf{y}_i), \mathbf{y}_i') - \nabla \hat{R}(\boldsymbol{\theta}) \right\|_2^2}{\left\| \nabla \hat{R}(\boldsymbol{\theta}) \right\|_2^2}.$$

