# OpenReview forum: "Analyzing the Sample Complexity of Self-Supervised Image Reconstruction Methods"
_NeurIPS.cc/2023/Conference — NeurIPS 2023 poster_

### Official Review · Reviewer_X8db · 2023-07-04

**Soundness:** 3 good
**Presentation:** 3 good
**Contribution:** 2 fair
**Rating:** 5
**Confidence:** 4

**Summary:**

The paper presents a theoretical analysis of the sample complexity of the problem of learning a linear denoiser with a self-supervised learning loss, and verifies the bounds on a series of experiments with linear denoising. The paper also presents an empirical study of the gap between self-supervised learning and supervised learning in the context of image reconstruction with neural networks (for denoising and compressive MRI problems).


**Strengths:**

- A theoretical bound illustrates the role of sample complexity in the setting of self-supervised learning for image denoising. A bound scaling as 1/N where N is the dataset size is presented.
- The gap between supervised and self-supervised learning is evaluated for various imaging tasks, showing that self-supervised losses which are unbiased estimators of the supervised loss can achieve a performance on par with supervised learning for large sample sizes.



**Weaknesses:**

There is little link between the theoretical analysis of linear denoisers and the empirical results on non-linear denoising and reconstruction with deep networks. It is not clear whether the theory developed in the linear case can explain the non-linear setting.
Moreover, some assumptions in the main theorem seem unrealistic in the context of deep learning, e.g. networks are trained on multiple epochs, not on a single pass as required by the theorem.


**Questions:**

Does the dimension of the signal set play a significant role in Theorem 1? While the dimension d appears in eq. 5, it doesn't seem to impact strongly the final bound, which is somehow surprising.

Why the paper doesn't analyze learning with a SURE-based loss? This should also be an unbiased estimator of the supervised loss for the denoising case.

**Limitations:**

The paper discusses the limitations related to the choice of an specific neural network. However, I think it would be good to include a discussion of the limitations of using a linear denoiser analysis to understand the dynamics of learning highly non-linear denoisers.

---

> ### Author Rebuttal · Authors · 2023-08-09
>
> Many thanks for the feedback. In the following we address the weaknesses and questions pointed out by the reviewer.
> - **Weakness and limitations; on the connection of theory and practice, and our theory pertaining to a linear estimator:** We think our theory and empirical results for the linear estimator and the empirical results for the neural networks are well connected in that the qualitative behavior found for a linear estimator theoretically is also observed empirically for neural networks.
> In particular, our theoretical results for denoising predict that the gap between noise2noise and supervised training depends on the noise variance $\sigma_e^2$ of the training targets and closes as the number of training examples $N$ increases, and that's exactly what we find empirically for neural networks as well.
> *Regarding the theory pertaining to a linear estimator:* In our theory section, we focus on a linear estimator as for the linear estimator we can make a precise statement, while for a complicated neural network we cannot. However, our result also applies to a more general setup, in particular to any estimator for which the loss function is strongly smooth and has a bounded stochastic gradient as formalized in the following Theorem, which we added along with proof and discussion to the appendix:
> > **Theorem 2.** *Consider the estimate $\mathbf{\theta} _ N$ obtained by running the SGM for $N$ iterations on the training set $\mathcal{D}$ with a decaying stepsize $\alpha_k = \frac{1}{c+k}$, where $c$ is a constant. Assume that the loss $h(\mathbf{\theta}) = ||f_{\mathbf{\theta}}(\mathbf{y}) - \mathbf{y}'||_2^2$ for any pair $(\mathbf{y},\mathbf{y}')$ is $Q$-strongly smooth and that the stochastic gradient is $(M,B)$-bounded. Then the expected generalization error, where expectation is over the random training set $\mathcal{D}$ obeys*
> $$
> \mathbb{E} [R(\mathbf{\theta} _ N) ] \leq R(\mathbf{\theta}^*) + \frac{Q}{2} \frac{1}{N-2} \frac{1}{m^2} (2M^2 e_0 + B^2).
> $$
> >
> The Theorem follows from the definition of strong smoothness and Lemma 1 in Appendix B and its interpretation is equivalent to the interpretation of Theorem 1 for the linear case: We get a rate of $1/N$ and the term associated with maximizing the self-supervised loss becomes larger in the noise variance $\sigma_e^2$ on the training targets since a larger noise variance requires a larger parameter $B$.
> - **Weakness part 2, regarding the assumption in Theorem 1 to consider a single pass of the stochastic gradient method:** With a single pass over the training set we already get an optimal risk bound (up to constants), and thus there seems little value in analyzing multiple passes. Analyzing a single pass of SGD over a training set is a standard technique in the analysis of SGD and is widely accepted, for two reasons: Often it is sufficient to get optimal bounds up to numerical constants (as in our setup), and moreover performing multiple passes significantly complicates the analysis since then we can't leverage independence of the samples as efficiently.
> - **Question part 1, the role of the signal dimension d in Theorem 1:** Good question, the signal dimension doesn't play a significant role in the theorem, since we scale the signal energy to be one in expectation, irrespectively of the signal dimension. The other energies are also scaled to be independent of the signal and ambient dimension. The signal dimension does, however, play a role in that it determines how much noise is filtered out (i.e., the factor $d \sigma_z^2/n$), this is the standard factor that we expect from subspace denoising.
> - **Question part 2, why do we not consider a SURE-based loss?** That is an excellent question. The SURE loss gives an unbiased estimate of the loss under additional assumptions, i.e., Gaussianity of the noise. We consider a class of self-supervised  noise2noise-like losses that is significantly more general in that it applies beyond Gaussian denoising to for example real-world camera denoising (see the pdf attached to our global response) and in that it generalizes to compressive sensing as discussed in the paper. We are interested in this more general class as it is much more widely applicable.
>
> We hope that our clarifications above address the reviewer's concerns and would appreciate it if the reviewer would consider raising their score. We are also happy to discuss further, thanks again for your comments.

---

> > ### Comment · Reviewer_X8db · 2023-08-14
> >
> > Many thanks for answering my questions. I have raised my score accordingly.

---

### Official Review · Reviewer_8mky · 2023-07-07

**Soundness:** 3 good
**Presentation:** 3 good
**Contribution:** 3 good
**Rating:** 5
**Confidence:** 4

**Summary:**

The work investigates the cost of self-supervised training by characterizing its sample complexity.

**Strengths:**

1. The paper is based on the given theory and carries out corresponding empirical research on self-supervised denoising and accelerated MRI.
2. The paper shows that a model trained with such self-supervised training is as good as the same model trained in a supervised fashion, but self-supervised training requires more examples than supervised training.
3. The paper shows that the performance gap between self-supervised and supervised training vanishes as a function of the training examples, at a problem-dependent rate, as predicted by the theory.


**Weaknesses:**

1. The main concern is that the theoretical approach of this paper seems to be similar to [1], just extending from supervised to self-supervised settings. The corresponding contribution of the theoretical approach should be further elucidated.
2. In the results reported by some previous self-supervised denoising works (e.g., Neighbor2Neighbor), Noise2Noise generally performed the same as supervised methods. But in this work, even with a lot of training data, there is still a gap between the two, what is the reason for this?
3. The paper is based on the setting of simple Gaussian noise. I would like to ask if the authors have done corresponding research or experiments on real-world RGB noise. Is this work still applicable to real-world situations?

[1] Scaling laws for deep learning based image reconstruction. ICLR 2023.

**Questions:**

Please see the weaknesses. I am willing to improve the score if the concerns are addressed well.

**Limitations:**

The limitations have been discussed in the paper.

---

> ### Author Rebuttal · Authors · 2023-08-09
>
> We thank the reviewer for the feedback. In the following we address the weaknesses in the order as pointed out by the reviewer.
> - **Weakness 1, 'the theoretical approach of this paper seems to be similar to [1]':** We would like to point out that the theoretical approach of this paper is different from that of [1] in that both the setup and the proof technique are substantially different. Our work analyzes the self-supervised noise2noise loss, while [1] considers a standard supervised loss. Also, [1] considers an early-stopped estimator while we do not. The proof technique used in [1] is different from ours. Ours is based on a convergence analysis of the stochastic gradient method, while [1] is based on analyzing the iterates of gradient descent.
> - **Weakness 2, noise2noise performing equivalently to supervised training in other works:** Yes, in some publications a network trained with a noise2noise loss performs as well as a network trained with a supervised loss. However this is misleading as it typically pertains to an unrealistic setup where the noise is resampled.
> Specifically, the original noise2noise paper [2] re-samples the noise on the noisy training targets in every training epoch, which requires the original image. Follow-up work like neighbor2neighbor [3] or noisier2noise [4] stuck to this approach. However, this is an unrealistic setup, since if we are given the ground truth images we can and should just train in a supervised manner.
> In our paper we consider the more realistic setup where we are given two noisy realizations of an image only. An important contribution of our work is to show for this setup, a gap between supervised and noise2noise training exists, which closes as the number of examples becomes large.
> - **Weakness 3, noise2noise for real-world image denoising beyond Gaussian noise:**
> That is an excellent point. Noise2noise like training for denoising is applicable beyond Gaussian noise. The condition formulated in Proposition 1 only requires the noise on the training targets to be uncorrelated with the residual.
> There are a variety of situations in practice where noise2noise like training is applicable. To demonstrate this point, we conducted additional experiments during the rebuttal period on real-world camera image denoising to empirically determine the performance gap between models trained in a self-supervised noise2noise and a supervised manner as a function of the number of training examples. Our results on denoising the raw images in the Smartphone Image Denoising Dataset (SIDD) [6] show that an initial gap of 2.6dB in PSNR for only 100 training patches reduces to 0.2dB for 100k patches. See the attached pdf for the full results that we'll include in the paper.
> There are other real-world settings where our results are applicable, e.g., to real fluorescence microscopy images noise, see the paper [5].
>
> We hope that our clarifications above address the reviewer's concerns. In particular we hope the reviewer's main concern, i.e., that the theory is similar to [1], has been addressed, and the secondary concern has been addressed with the new simulations on additional real-world-simulations for denoising. If this is the case, we would appreciate it if the reviewer would consider raising their score. We are very happy to discuss further, thanks again for your comments.
>
> [2] Lehtinen et al. Noise2Noise: Learning Image Restoration without Clean Data. ICML 2018.
> [3] Huang et al. Neighbor2Neighbor: Self-Supervised Denoising From Single Noisy Images. CVPR 2021.
> [4] Moran et al. Noisier2Noise: Learning to Denoise From Unpaired Noisy Data. CVPR 2020.
> [5] Zhang et al. A Poisson-Gaussian Denoising Dataset with Real Fluorescene Microscopy Images. CVPR 2019.
> [6] Abdelhamed et al. A High-Quality Denoising Dataset for Smartphone Cameras. CVPR 2018.

---

> > ### Author Response · Authors · 2023-08-17
> > **Checking in**
> >
> > Thanks a lot again for your review and feedback. We hope we have addressed your concerns. Please let us know if you have any remaining concerns and questions.

---

> > > ### Comment · Reviewer_8mky · 2023-08-18
> > >
> > > Thanks for answering the questions. The concerns have been addressed. I have raised my rating.

---

### Official Review · Reviewer_RTiS · 2023-07-12

**Soundness:** 3 good
**Presentation:** 4 excellent
**Contribution:** 2 fair
**Rating:** 7
**Confidence:** 3

**Summary:**

The paper is an study on the sample complexity for image reconstruction in two types of methods, self-supervised and supervised. The authors studies the risk bounds for the case of self-supervised methods. They then evaluate the convergance rates in numerical and empirical experiments for two problems, denoising and compressive sensing. The authors conclude that the convergence rates are similar in the two cases of unsupervised and supervised scenarios. However, the self-supervised approach requires more iterations to reach a similar performance.

**Strengths:**

1) The authors theoretically study and find specific risk bounds for using the self-supervised method.

2) The authors study the sample complexity empiracally by considering a range of number of parameters for the network and a range of training set sizes.

**Weaknesses:**

1) For many of the empirical experiments, the authors only report the best out of multiple runs. Illustrating the mean and some measure of variance among multiple runs gives a more complete picture rather than just the best case. Otherwise, the readers would wonder how reliable it is to do a single run of the self-supervised approach compared to the supervised approach.

**Questions:**

1) In appendix 1, in the first line, I think the sign of the last term, $e$, should be negative, based on the given definition for $y\prime$. This results in changing the sign of a few terms in following lines. But the conclusion still holds.

2) Since there is an assumption that the noise distributions for training and inference to be the same in the case of compressed sensing, is it fair to call it self-supervised?

3) As the authors also point to in their limitations segment, the experiments are limited to U-net like design for the architectures. However, they mention they do not expect using different designs would change the qualitative results. Could they ellaborate on the intuition behind this expectation?

- Typo:
    - Line 511: missing reference

**Limitations:**

The title and the claims in the paper point to image reconstruction in general. However, the experiments are only limited to denoising and compressed sensing. The behavior might be very different for some other image reconstruction tasks such as image inpainting or super-resolution. This limitation should be more pronounced in the claims.

---

> ### Author Rebuttal · Authors · 2023-08-09
>
> We thank the reviewer for their feedback and their positive evaluation of our work. In the following we address the weakness and questions pointed out by the reviewer.
> - **Weakness 1, reporting only the best runs:** For all empirical results on denoising and compressive sensing presented in Figure 2,4,5, the Figures show all conducted runs, not only the best ones. The figures show that as the number of training examples increases the variance of the runs decreases, which is why for the largest training set sizes we only conduct a single run. Instead of drawing the performance curves in Figure 2,4,5 based on the best performing models, we could also use the mean performance, which would shift all curves a little bit downwards but would not affect our findings.
> - **Question 1:** Many thanks for pointing out the typo, we fixed it.
> - **Question 2, is it fair to call the compressive sensing setup self-supervised:** Yes, we believe it is fair to call the accelerated MRI compressive sensing setup self-supervised since we do not have access to fully-sampled data during training. Specifically, during training, we only assume that we have access to 1/3 of all possible measurements, which we then split into a network input with an undersampling factor 1/4 and a corresponding training target containing the remaining measurements plus some overlap. At inference, we assume access to only 1/4 of all possible measurements. Contrary, a supervised method assumes access to fully-sampled data at training time.
> - **Question 3, why we expect the choice of network architecture not to affect our qualitative results:** Our result that the performance of a model trained in a noise2noise self-supervised way approaches the performance of a model trained in a supervised way, is based on how well two different loss functions approximate the risk, and thus the qualitative findings do not depend on the explicit network architecture. This can be also seen from Proposition 1 and 2 in which we formulate the conditions under which this result holds for denoising and compressive sensing and which do not depend on the particular choice of the network $f _ {\mathbf{\theta}}$. Motivated by your question we added a reference to the Propositions in the limitations segment to make this point more clear.
> - **Limitations, regarding the generality of our claims:** As pointed out by the reviewer, our results are for a class of self-supervised methods based on constructing unbiased estimates of the gradients of the supervised loss and pertain to denoising and accelerated MRI. There are other problems where such self-supervised losses can be constructed, for example for CT imaging and for some cryo-EM setups, and for those setups the qualitative results from our paper also apply. However, we fully agree that the results and statements in our paper pertain to denoising and accelerated MRI. Also, there are many problems where two independently obtained measurements cannot be easily obtained, and then the class of methods discussed in this paper does not apply. We made both points more clear in the limitations section as suggested. Thanks!
>
> Thanks again for your review, and please let us know if you have any other questions.

---

> > ### Comment · Reviewer_RTiS · 2023-08-15
> > **Response to the authors**
> >
> > Thank you for your detailed response to the reviewers' comments on your paper.

---

### Author Rebuttal · Authors · 2023-08-09

Dear reviewers,

Attached is a pdf containing experimental results on real-world camera image denoising as discussed in our response to reviewer 8mky (weakness 3), who asked if our results hold for real-world noise beyond the Gaussian setup studied so far in our paper.
The results demonstrate how the performance of models trained in a noise2noise self-supervised way approaches the performance of models trained in a supervised way as a function of the number of training examples also for real-world noise analogously to our previous results for Gaussian denoising.

We hope that you find the additional material helpful and are happy to discuss further.

---

### Decision · Program_Chairs · 2023-09-21

**Decision:**

Accept (poster)

**Comment:**

The paper presents theoretical and empirical analysis of self-supervised inverse problems for image reconstruction. The three reviewers agreed the paper was above acceptance threshold. The result that supervised and unsupervised denoising performs asymptotically similarly is of general interest to the ML community and the experiments and theory was well developed and should be of broad interest to ML researchers. Therefore, the paper would be a useful contribution to the proceedings.